# Conveying tactile object characteristics through customized intracortical microstimulation of the human somatosensory cortex

Ceci Verbaarschot [1,2,3], Vahagn Karapetyan[1,4], Charles M. Greenspon [5], Michael L. Boninger [1,3,4], Sliman J. Bensmaia[5,6], Bettina Sorger [2] & Robert A. Gaunt [1,3,4] ✉

Microstimulation of the somatosensory cortex can evoke tactile percepts in people with spinal cord injury, providing a means to restore touch. While location and intensity can be reliably conveyed, two issues that prevent creating more complex naturalistic sensations are a lack of methods to effectively scan the large stimulus parameter space and difficulties with assessing percept quality. Here, we address both challenges with an experimental paradigm that enables three male individuals with tetraplegia to control their stimulation parameters in a blinded fashion to create sensations for different virtual objects. Using this method, participants can reliably create object-specific sensations and report vivid object-appropriate characteristics. Moreover, both linear classifiers and participants can match stimulus profiles with their respective objects significantly above chance without any visual cues. Confusion between two sensations increases as the associated objects share more tactile characteristics. We conclude that while visual information contributes to the experience of the artificially evoked sensations, microstimulation in the somatosensory cortex itself can evoke intuitive percepts with a variety of tactile properties. This self-guided stimulation approach may be used to effectively characterize percepts from future stimulation paradigms.

The ultimate goal of prosthetics research is to create an artificial limb that effortlessly replaces lost function by seamlessly integrating the device with a person's existing sensorimotor system. Tactile feedback is a key element to achieve this goal[1–5], and a much-desired feature amongst prosthesis users[6,7]. Intracortical microstimulation (ICMS) of the somatosensory cortex can evoke localized sensations on a person's paralyzed or insensate hand[8–10] and is a promising way to provide this feedback. ICMS delivers tactile information directly to the brain, making it an attractive option for people with spinal cord injury or high-level amputation. How to best translate tactile object characteristics into stimulation patterns is non-trivial due to a limited understanding of the neural processing of touch, the complexity of the

[1]Rehab Neural Engineering Labs, University of Pittsburgh, Pittsburgh, PA, USA. [2]Psychology and Neuroscience, Maastricht University, Maastricht, Netherlands. [3]Department of Physical Medicine and Rehabilitation, University of Pittsburgh, Pittsburgh, PA, USA. [4]Department of Bioengineering, University of Pittsburgh, Pittsburgh, PA, USA. [5]Department of Organismal Biology and Anatomy, University of Chicago, Chicago, IL, USA. [6]Deceased: Sliman J. Bensmaia. ✉e-mail: rag53@pitt.edu

stimulation parameter space, and hardware restrictions that limit our ability to replicate naturally occurring neural responses using ICMS[11,12]. Moreover, measuring artificial percepts can be challenging due to the inherent complexity of capturing an experienced sensory quality into objective measurements. We cannot guarantee that two experiences are similar, even when they are described with exactly the same words. Sensation reports can be difficult to interpret and prone to bias. For example, if a person is asked how much "pressure" they felt on their hand during a stimulus, a participant may interpret a "hard" sensation as feeling like high pressure and a "soft" sensation as low pressure. However, they may not have described the sensation as "pressure" at all in the absence of this question.

So far, most studies focusing on the psychophysics of ICMS have manipulated one stimulation parameter at a time. The majority of these studies have been conducted in non-human primates that cannot verbalize their experienced sensations[13–15]. Human participants typically describe ICMS-evoked sensations as "possibly natural"[10], reporting a wide variety of naturalistic qualities such as "pressure", "press", "tap", "warm", "squeeze", "pinch", "vibration", "blowing", and "goosebumps"[8,10,16]. In addition, some more artificial qualities such as "pins and needles", "tingle" and "electrical" have been reported[8,16]. The perceived qualities depend on the stimulation amplitude, frequency and (multi-electrode) location(s)[8,10,15,17,18]. Many combinations of stimulus parameters have yet to be explored, in part because the number of combinations grows exponentially as parameters are added. Evaluating the resulting sensations one by one using verbal reports or survey ratings may not be effective due to the theoretical challenges described above, and their time consuming and repetitive nature. This drives a need for more efficient methods to explore the quality of ICMS-evoked sensations than the commonly applied stimulate-and-report task[8,10,16].

Here, we present an interface that three individuals with tetraplegia used to design their own sensations. Participants controlled stimulation presentation by "touching" an object displayed on a tablet (Fig. 1a), creating a more realistic experimental context compared to previous research. In this case, the experienced sensations are the result of targeted explorative movements rather than experimenter driven stimulation without a meaningful context. Moreover, through the self-paced scanning of a pre-selected parameter space (Fig. 1b), participants could adjust stimulation parameters without having to complete a sensory survey after each new parameter[8,10,19,20], providing an efficient and engaging experimental paradigm. Participants controlled the stimulation amplitude, frequency, "biomimetic factor" and "drag", while remaining blinded to the parameters themselves[21] (Fig. 1b, c). Changes in amplitude and frequency have been shown to influence the perceived quality and intensity of the evoked sensations[8,10,15,18]. Further, most ICMS trains have commonly comprised fixed amplitudes and frequencies, or a stimulus amplitude that varies linearly based on an input variable[8,10,22]. However, neural responses in somatosensory cortex are dominated by a large burst of activity at the onset and offset of object contact, while sustained contact generates relatively low-level activity[15]. Using such transients ("biomimetic factor") may evoke more intuitive, pleasant, or familiar sensations as it seeks to mimic patterns of natural brain activity[17]. Lastly, to simulate the experience of the participants' hand moving across an object, we stimulated three electrodes sequentially; each electrode evoked a sensation on an adjacent area on the hand. To create diffuse temporal transitions between consecutive electrodes, we created a "drag" parameter that controlled the degree of stimulation overlap between electrodes (Fig. 1c). With these methods, we investigated whether more complex stimulation trains could evoke intuitive tactile object characteristics and identified specific stimulation parameters that best matched different objects.

By putting participants in direct control of their own brain stimulation, we hoped to keep people engaged in exploring a large parameter space to develop personalized object-specific stimulus parameters. Our results show that participants felt they could reliably create stimulus-driven sensations that had object-appropriate characteristics for the visually presented objects. While the specific stimulus parameters that people chose for each object showed some overlap, they were nevertheless able to identify the correct object when stimulation was delivered without any visual context significantly above chance. Further, the performance of the participants in this perceptual classification task was very similar to the performance of linear classifiers trained to identify objects based on the stimulation parameters alone. Finally, objects that had more similar natural properties were more likely to be confused with each other without the visual context, suggesting that participants were selecting stimulation parameters that represented real objects properties. Given the large (and blinded) parameter space that was included in this study, it is promising that even in this challenging environment, participants were able to create distinct object sensations. We conclude that more complex stimuli, like those presented here, unlock a greater perceptual space that may allow people to distinguish artificially perceived objects with increased precision and intuition.

## Results

Three participants with tetraplegia (P2, P3, C1) designed their own artificial tactile sensations to represent interactions with a cat, apple, towel, piece of toast, and key. These objects were chosen because they spanned a range of tactile dimensions (compliance, temperature, friction, moisture, macro texture, micro texture, pleasantness, familiarity), as judged by 34 people with intact somatosensation (Fig. S1a) in an online survey (Fig. S1b). The participants used residual function in their left hand to interact with a tablet interface that generated "touch" sensations on the palmer surface of their right hand (Fig. 1a). Whenever participants touched a virtual object with their left hand, stimulation trains were delivered through microelectrode arrays implanted into Brodmann's area 1 of the left hemisphere. Three electrodes were driven by contact with different regions of the object, such that the sensations 'moved' across their right hand as they moved across the object (Fig. 1b). In addition, participants used their left hand to control the position of cursors in two rectangular spaces on a tablet. Using these cursors, participants controlled the stimulus amplitude, frequency, biomimetic factor, and drag parameters (Fig. 1b, c). The parameters were randomly assigned to each of the four axes at the start of a trial, blinding participants to which axis controlled which parameter[21].

During the first phase ("object-sensation mapping task", see Methods), participants repeatedly selected stimulation parameters that best represented each object. At the end of each trial, participants were asked to rate their satisfaction with the created sensation. In the second phase, we assessed the discriminability of the created sensations by having participants complete a "replay task" at the end of each session (Fig. 1d). During this task, stimulation trains that were created in the first phase were replayed without the corresponding object image. Participants were then asked to select the object that best matched the experienced sensation. In addition to these same day replay tests, two additional sessions used the replay task to test the discriminability and stability of a selection of sensations that were created across different days. For an overview of the data collection timeline, see Fig. S2.

### Stimulation parameter selections were object specific

All participants described vivid memories and ideas of what the target sensations in this experiment should feel like. They expressed no doubt or confusion about what sensation they wanted to achieve for the various object interactions. To validate whether the stimulus patterns generated using the interface could produce a variety of perceptual qualities, we first investigated how participants explored the

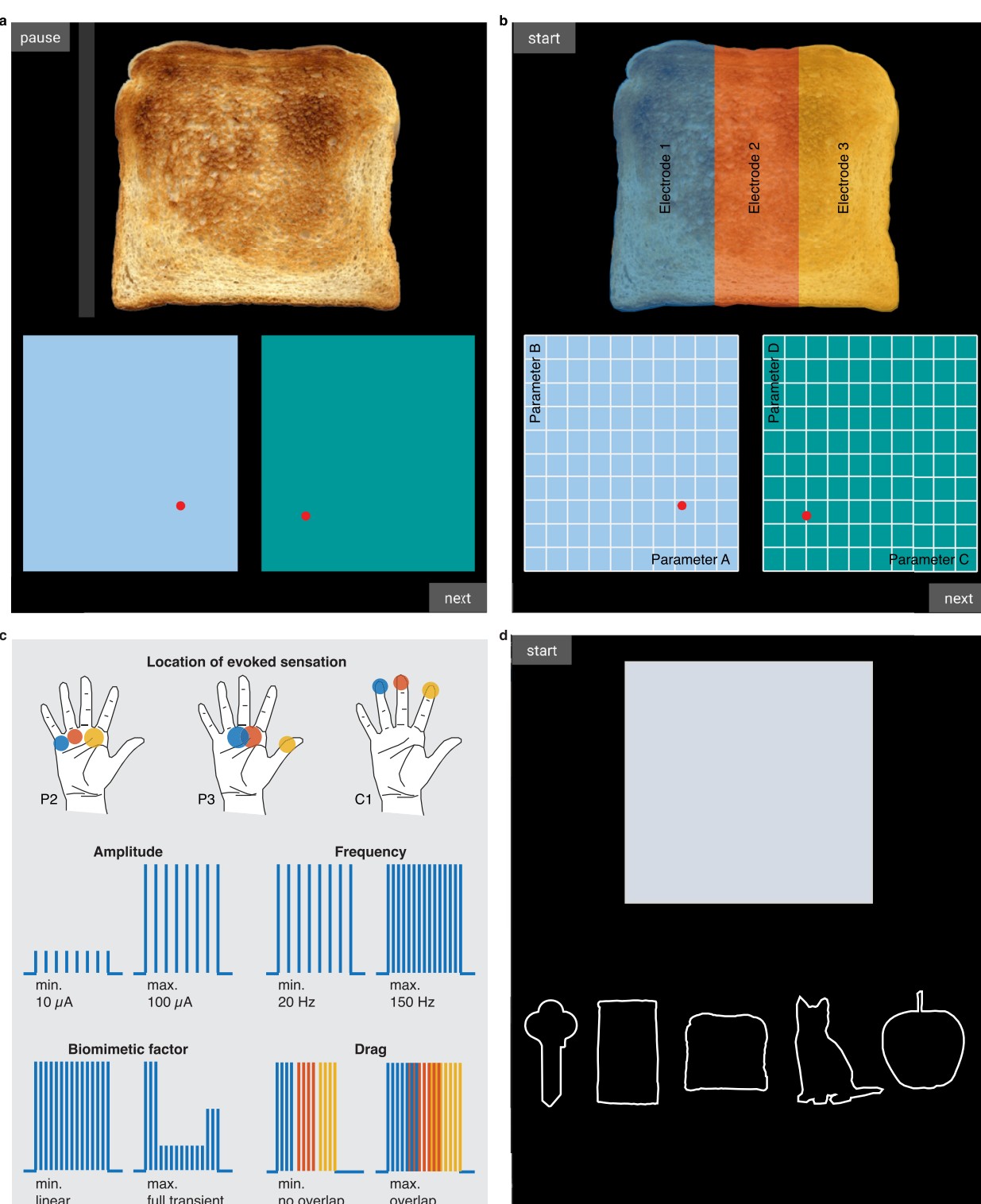

stimulus parameter space during individual trials. Participants P2 and C1 broadly explored the space, but also identified "hotspots" in individual trials where they focused their attention (Fig. 2a). The presence of hotspots suggests that certain parameter combinations evoked more appropriate sensations for a given object than others (Fig. S3). Participants P2 and C1 had similar trial completion times (Fig. 2b), explored roughly equal parts of the parameter space (Fig. 2c) and received comparable amounts of stimulation during a trial (Fig. 2d). In contrast to P2 and C1, participant P3 spent significantly less time on

each trial, exploring less of the parameter space and receiving less stimulation (Fig. 2b–d).

Although the variance in participant's parameter selections was high, the resulting stimulus profiles occupied distinct locations in the parameter space, depending on their target object (Fig. 3a). The separation between object-specific stimulus profiles was less clear in participant P3 compared to P2 and C1, likely due to his shortened exploration times (Fig. 2b–d). Nevertheless, the parameter selections of sensations that participants recognized consistently correctly in the

**Fig. 1 | Tablet interface. a** Screenshot of the object-sensation mapping task. By changing the positions of the cursors in the left and right rectangles, the participant could adjust four stimulation parameters. Touching the object ('piece of toast' in this example) generated stimulation trains in real-time. Participants manually touched the digital object or used an automated cursor by selecting the "start"/"pause" button in the top left corner, causing the bar to move across the object every 6.5 s. The toast image was made by Rainer Zenz and sourced from Commons.wikimedia.org (license: creativecommons.org/licenses/by-sa/3.0). No changes were made to the image. No changes were made to the image. **b** At the start of a trial, the four stimulation parameters (amplitude, frequency, biomimetic factor, drag) were randomly assigned to the A–D locations on the *X*- and *Y*-axes of the two rectangles. In addition, parameter values were randomly assigned to increase or decrease along the ten possible levels for each axis. Depending on where the participant touched the object, one of three stimulation electrodes (shown in different colors) was stimulated with the settings defined by the location of the cursors, evoking localized sensations on the palmar side of their right hand. **c** The exact locations of the evoked sensations by each stimulation electrode (shown in different colors on the hand images) were different for each participant, depending on the locations of the stimulation arrays in their somatosensory cortex. For more details on the exact implant locations for each participant, see Fig. 1 in ref. 24. The remainder of the figure shows a schematic illustration of the stimulation parameters at their minimum (min.) and maximum (max.) levels: amplitude, frequency, biomimetic factor, and drag. Each parameter controlled the stimulation of three individual electrodes and could be changed across 10 levels, i.e., there were 10 unique values per parameter. **d** Replay task. When the participant touched the rectangle, a previously chosen set of stimulation parameters was replayed, evoking a sensation across the participants' right hand. The participant was asked to indicate which object best matched the experienced sensation.

replay task, showed greater similarity within the same object category than across different objects (P2: $p = 0.005 \times 10^{-3}$, one sided Wilcoxon rank sum, $\alpha = 0.05$, $z = -4.428$; C1: $p = 0.004 \times 10^{-1}$, $z = -3.362$, Fig. 3b), albeit non-significantly for P3 ($p = 0.084$, $z = -1.379$, Fig. 3b). The closer a stimulus set was to the median across all successfully recognized stimulus profiles of a certain object, the more likely the participant was to correctly identify the sensation as belonging to that object (P2: $p = 0.003^{-2}$, one sided Wilcoxon rank sum $\alpha = 0.05$, $z = -4.146$; P3: $p = -0.007^{-1}$, $z = -3.396$; C1: $p = 0.011$, $z = -2.532$, Fig. 3c). From these successfully recognizable stimulus profiles, a rough mapping from stimulus parameters to sensory object characteristics could be inferred (Fig. 3d).

The separability of object-specific stimulus profiles was driven by combinations of multiple parameters rather than individual parameters. To assess the distinctiveness of combined parameter settings, we trained a linear discriminant analysis (LDA) classifier to identify which of the five objects a specific set of parameters belonged to using tenfold cross validation. Across all five objects, the stimulation parameters predicted the object identity with an accuracy of $34 \pm 12\%$ for participant P2 ($p = 0.002$, one-sided permutation test with 1000 permutations, $\alpha = 0.05$, Fig. 4a) and $37 \pm 15\%$ for C1 ($p = 0.002$, Fig. 4a). At an accuracy of $21 \pm 13\%$, the stimulation parameters chosen by P3 predicted the associated object no better than chance ($p = 0.341$, Fig. 4a). By repeating this analysis using each possible combination of stimulus parameters, we confirmed that the best classification result was achieved using all four parameters rather than a single parameter or pair of parameters (Figs. 4b and S5a–c). There were only a few instances where there were statistically significant differences in the distributions of individual stimulation parameters for different objects (Table S1). Specifically, in the amplitude of both P2 ($p = 0.001^{-1}$, two-sided Kruskal–Wallis test Bonferroni corrected at $\alpha = 0.006$, $\chi2(4) = 23.16$, Fig. S4a) and C1 ($p = 0.002^{-1}$, $\chi^2(4) = 21.86$, Fig. S4b), and in the biomimetic factor ($p = 0.003$, $\chi2(4) = 16.13$, Fig. S4a) and drag ($p = 0.004$, $\chi2(4) = 15.50$, Fig. S4a) of P2. Moreover, classification performance could not be explained simply by differences in the total charge delivered per electrode for each object (Fig. S6).

The object-specific stimulation parameters that drove LDA classifier performance also evoked distinct percepts that the participants could use to complete the replay task (Fig. 1d). Both participants P2 and C1 could use a given set of stimulus parameters to identify the corresponding object significantly above chance; P2 correctly identified the matching object with a mean accuracy of $36 \pm 16\%$ (one-sided permutation test, 1000 permutations, $\alpha = 0.05$, $p = 0.001$ Fig. 4a) per session and C1 did so with an accuracy of $37 \pm 13\%$ ($p = 0.001$ Fig. 4a). As expected, participant P3 performed around chance with a mean accuracy of $22 \pm 9\%$ ($p = 0.051$, Fig. 4a). For some objects, the participants created more distinctive stimulation parameter sets than others. This was reflected in the ability of LDA classifiers to accurately predict certain objects as well as the ability of the participants to recall the correct object in the replay task (Figs. 4c and S5d–f). We further investigated the distinctiveness of the created sensations by determining the LDA and participant performances on identifying each sensation of a specific target object from all others. At least three out of five object-sensation categories could be recognized significantly above chance by participants P2 (cat, key, towel toast, Fig. 4d) and C1 (cat, apple, towel, Fig. 4d). Even P3 performed significantly above chance on a single category of sensations (key). In general, we observed that the more distinct the chosen stimulus profile was, the easier participants could distinguish them in the replay task (Pearson's $r(18) = -0.45$, $p = 0.045$, Fig. 4e).

The significant performances of the LDA classifier for participants P2 and C1 suggest that there is some consistency in the selected object-specific parameters across sessions. To investigate the perceptual stability of the created sensations, we conducted two additional sessions of the replay task three (P2, C1) to seventeen (P3) weeks after finishing data collection. This time, we used three stimulus parameter sets per object that were created across previous sessions. All participants assigned two (P2, P3: cat, key) to three (C1: cat, key, towel) out 15 tested parameter sets consistently to the correct object. However, their overall replay performance did not rise above chance level (P2–22%, P3–19%, C1–25%). This suggests that although there was some consistency in the parameter sets of individual objects over time, their session-to-session variations influenced their perceptual experience. Indeed, the object-specific variations in stimulus parameters were found to be significantly larger across than within sessions ($p = 0.002$, one-sided Kruskal–Wallis test, $\alpha = 0.05$, $\chi^2(1) = 9.95$, Fig. S7).

### Participants reported vivid and object-appropriate percepts

The participants described their sensory experiences at the end of each object-sensation mapping trial. These reports contained vivid object-appropriate sensory experiences. For example, P3 described the sensation for touching an apple as, "I found the perfect apple. It felt *light* but also *smooth*, *curved* and a little bit of *cool* and *wet*.". Participant C1 described a cat sensation as, "very *light touch*, just *like petting a cat*. *Smooth silkiness* on fingertips. *Resistance* of cat. Has that *oily sensation*. It even has a sort of *warmth* to it." Participant P2 described a towel sensation as, "It is *tingly*, but not quite *tappy*, but also not quite solid, reminds me of *roughness*." Overall, participants described characteristic object qualities like "smooth", "warm", "dry" and "rough" in their sensation reports. These words are distinct from those used in previous reports of standard microstimulation trains[8,10,23] and provide evidence of the experience of object-appropriate tactile characteristics in response to more complex customized microstimulation trains (Fig. 5).

In addition, we asked participants to rate their satisfaction with the artificial sensations that they had created. Median normalized satisfaction scores were 68% for participant P2, 85% for P3 and 91% for C1 (Fig. 6a), indicating that participant P2 was the least satisfied and

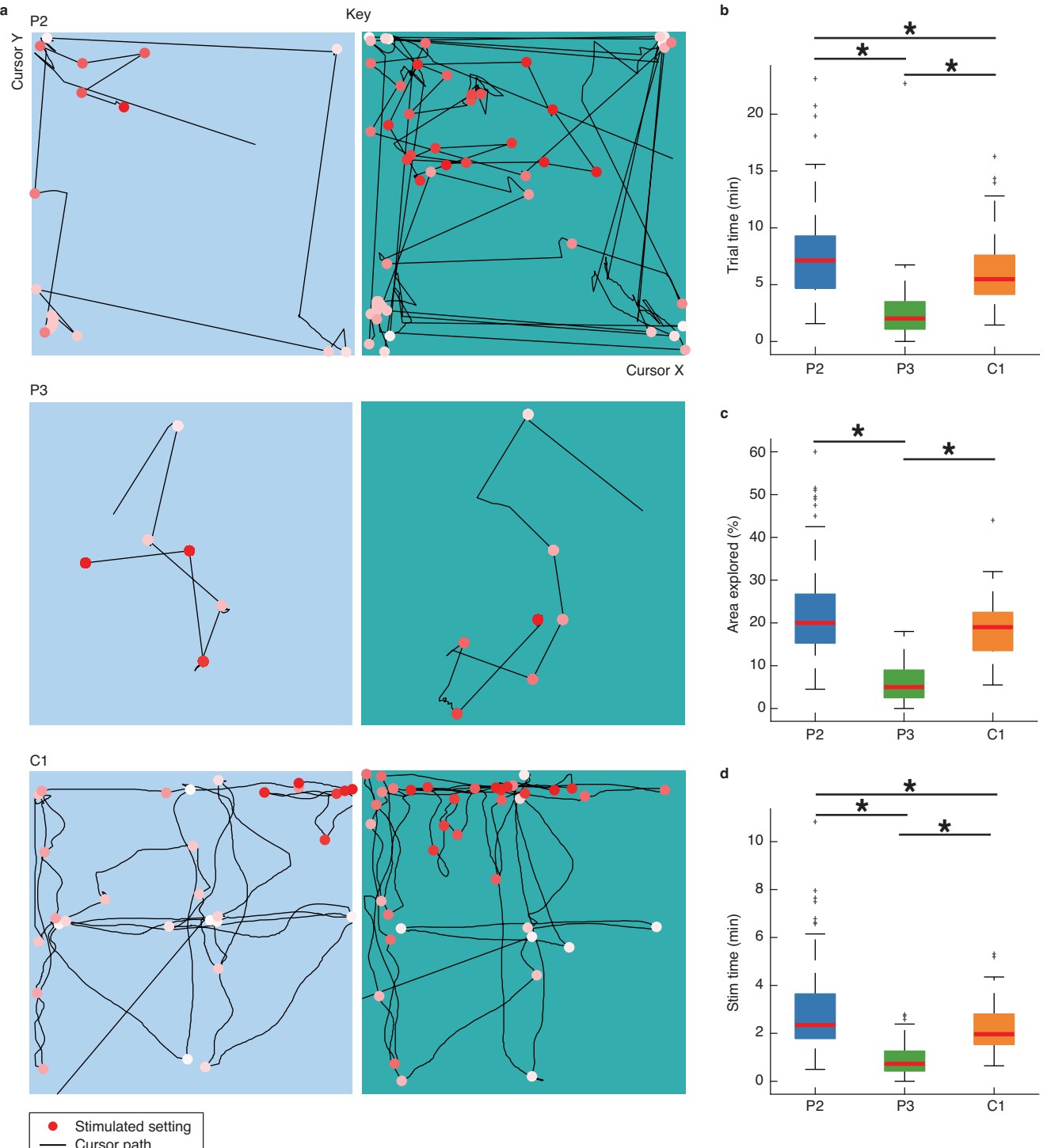

**Fig. 2 | Exploration behavior using the interface.** A total of 152 (P2), 103 (P2), and 104 (C1) trials were collected. Source data are provided as a Source Data file. **a** Participant behavior during an example object-sensation mapping trial for a key. Each dot indicates a stimulus parameter set during this single trial, changing from white (start trial) to red (end trial) as time progressed. The lines show the exact cursor movements. P2 explored the corners of each rectangle first followed by a more refined search starting from the most applicable corner; C1 started from the middle of each rectangle and tested systematic changes in sensation as he moved away from the middle; P2 did not report a clear search strategy. **b** Trial completion times for each participant. At $6.0 \pm 2.5$ min compared to $7.3 \pm 3.4$ min, participant C1 was slightly faster than P2 ($p = 0.001$, two-sided Wilcoxon rank sum, Bonferroni corrected at $\alpha = 0.008$, $z = 3.22$). At $2.4 \pm 2.4$ min, P3 took significantly less time per trial compared to both P2 ($p = 0.014^{-27}$, $z = 11.30$) and C1 ($p = 0.026^{-18}$, $z = -9.23$).

**c** Percentage of the parameter space explored per trial across all participants. At $19 \pm 7\%$ compared to $22 \pm 11\%$, participant C1 explored a similar amount of the parameters space as P2 ($p = 0.029$, two-sided Wilcoxon rank sum, $z = 2.18$). At $6 \pm 5\%$, participant P3 explored much less than both P2 ($p = 0.042^{-32}$, $z = 12.18$) and C1 ($p = 0.051^{-26}$, $z = -10.97$). **d** Total stimulation (stim) time within each trial for all participants. At $2.1 \pm 0.6$ compared to $2.5 \pm 1.4$ min, participant C1 had slightly less stimulation than P2 ($p = 0.004$, two-sided Wilcoxon rank sum, Bonferroni corrected at $\alpha = 0.008$, $z = 2.87$). At $54 \pm 40$ s, participant P3 had much less stimulation compared to both P2 ($p = 0.044^{-25}$, $z = 10.78$) and C1 ($p = 0.011^{-18}$, $z = -9.32$). Figures (**b**–**d**) show the median (central marker) and 25th and 75th percentiles (box) of the data, the limits of the data range (whiskers), and the outliers (+ marker). Significant differences are indicated with an asterisk (*).

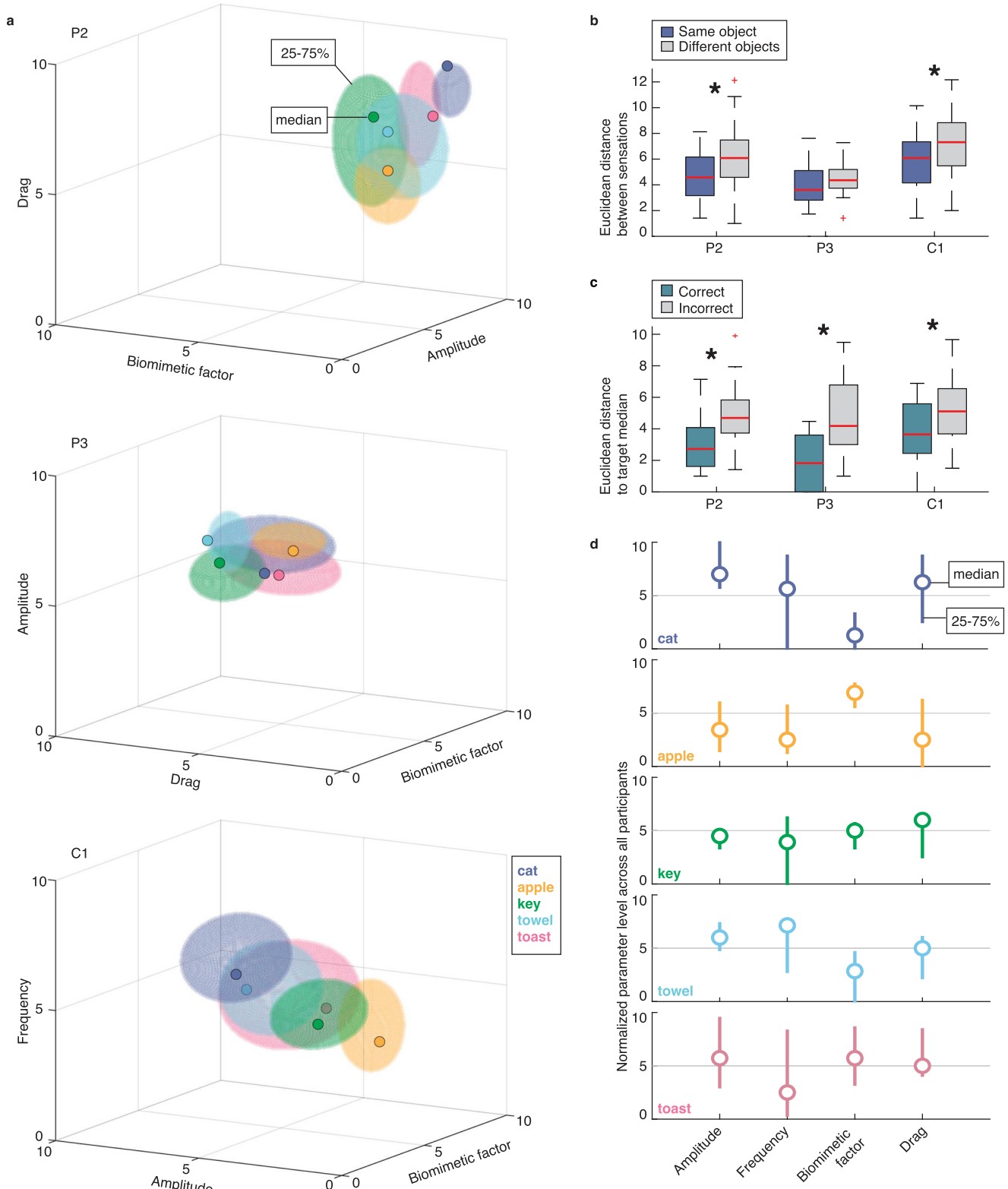

participant P3 the most satisfied with how well the ICMS-driven sensations represented the objects (Fig. 6a). There was a significant difference in the satisfaction scores between objects ($p = 0.001$, two-sided Kruskal–Wallis test, $\alpha = 0.025$, $\chi^2(4) = 18.26$, Fig. 6a), which was driven by a difference in cats and keys ($p = 0.005^{-1}$, two-sided Kolmogorov–Smirnov test Bonferroni corrected at $\alpha = 0.005$, $D = 0.33$, Fig. 6b). Although participants could evoke several satisfactory key-sensations, they were on average less happy with these

sensations compared to cat-sensations. No differences were found between the other object sensations.

During the replay task, we asked participants how certain they were of their decision. Both P3 and C1 were very certain of their answers (92%–100%), regardless of whether they were correct (Fig. 6c); there was no significant difference in certainty between correct and incorrect trials (P3, $p = 0.861$, Wilcoxon rank sum, $\alpha = 0.025$, $z = -1.07$; C1, $p = 0.114$, $z = 1.21$). In contrast, with a median of 63% for correct and

**Fig. 3 | Chosen stimulation parameter settings per object.** Source data are provided as a Source Data file. **a** Median and 25–75 percentiles of the selected stimulation amplitude, frequency and biomimetic factor for each participant. Different colors represent different objects. Only trials that had a normalized satisfaction score of at least 50 out of 100 were included, collected across 22 (P2, top) or 10 test days (P3, middle; C1, bottom). **b** Euclidean distance across all stimulation parameters of the same object vs. different objects for each participant. In (**b**, **c**), the median (central mark), 25th and 75th percentiles (box), data range (whiskers), and outliers (+ marker) are shown. Only sensations that were correctly recognized as belonging to the target object in the replay task were included. For participant P2 ($p = 0.005 \times 10^{-3}$, one sided Wilcoxon rank sum, $\alpha = 0.05$, $z = -4.428$, 51 same vs. 208 different samples) and C1 ($p = 0.004 \times 10^{-1}$, $z = -3.362$, 59 same vs. 195 different samples), the chosen stimulation parameters belonging to the same objects were significantly more similar to each other than those chosen for different objects.

This was not true for P3 ($p = 0.084$, $z = -1.379$, 13 same vs. 35 different samples). **c** Euclidean distance between each individual sensation parameter set and the median sensation parameters of correctly identified sensations in the replay task. We observe that for each participant, sensations that are closer to the median target sensation are significantly more likely to be correctly identified in the replay task (P2: $p = 0.003^{-2}$, one-sided Wilcoxon rank sum $\alpha = 0.05$, $z = -4.146$, 24 correct and 61 incorrect trials; P3: $p = -0.007^{-1}$, $z = -3.396$, 10 correct and 58 incorrect trials; C1: $p = 0.011$, $z = -2.532$, 24 correct and 71 incorrect trials). **d** Normalized median stimulation parameters selected across all participants for each object in the object-sensation mapping task. Thick lines indicate the 25–75 percentiles of the data. Only sensations that were correctly recognized as belonging to the target object in the replay task were included (14 cat, 14 apple, 10 key, 13 towel and 7 toast sensations).

51% for incorrect selections, P2 was significantly more certain of his answer when he was correct ($p = 0.003$, $z = 2.73$, Fig. 6c). Despite the lack of correlation between P3 and C1's certainty ratings and replay accuracies, these results suggest that participants experience subjectively distinct sensations in response to their object-specific parameter selections, even in absence of a visual context.

Lastly, participants completed a short survey on their overall experience with the interface. Participants reported that they liked the graphical interface, felt interested, involved, and drawn into the experimental task, found it fun and rewarding to do, and did not find the task frustrating, demanding, or confusing (Fig. S8). These results, together with the significant performances on the replay task, demonstrate that the tablet interface was an effective means to create vivid and object-appropriate sensations.

### Artificial percepts reflect naturalistic tactile features

To assess whether the quality of our participants' artificial sensations reflected the natural tactile characteristics of objects, we compared them to reports of object qualities provided by people with intact somatosensation. Using an online survey (Fig. S1b), 34 intact participants rated the compliance, temperature, friction, moisture, microstructure, and macrostructure of the objects included in this study. According to these survey results, the five objects in this study had a range of tactile properties that varied across multiple dimensions (Fig. S1a). The stimulus parameters that our participants selected correlated with these expected tactile characteristics; the more tactile characteristics two objects shared, the more likely they were confused by our participants in the replay task (Pearson's r(28) = 0.30, $p = 0.104$, Fig. 7a). Specifically, significant differences in compliance ($p = 0.001^{-13}$, one-sided Wilcoxon signed rank test, Bonferroni corrected at $\alpha = 0.013$, $z = -8.181$) and temperature ($p = 0.003^{-21}$, $z = -10.104$) explained most of the observed variance between different object groups (Figs. 7b and S9). For example, soft (i.e., cat, towel) and hard (i.e., apple, key, toast) object sensations were much more like sensations in that same compliance category than the opposite one. As such, a rough mapping could be identified between stimulus parameters and different levels of perceived object compliance and temperature (Fig. 7c).

To further assess the correlation between stimulus profiles and expected tactile object characteristics, individual LDA classifiers were trained to make predictions on different levels of object compliance, temperature, friction/texture and macro structure based on sets of stimulus parameters. To do so, the data was relabeled using the quality survey results. For example, for the "compliance" analysis, soft (cat, towel) and hard (apples, keys, toast) objects were grouped into two different groups. An LDA classifier was then re-trained to predict these tactile characteristics rather than object identities. This analysis confirmed the results above: for both P2 and C1, binary levels of object compliance (P2: $63 \pm 12\%$ accuracy, $p = 0.010$, permutation test with 1000 permutations, $\alpha = 0.025$; C1: $71 \pm 14\%$ accuracy, $p = 0.002$, Fig. 7d)

and temperature (P2: $72 \pm 11\%$, $p = 0.001$ C1: $76 \pm 13\%$, $p = 0.001$ Fig. 7e) could be significantly predicted. LDA classifier performance was not significant for the other tactile characteristics (Table S2).

Similar results were obtained by re-assessing the participant's replay performance using the same compliance and temperature labels as used for the LDA classifications (Fig. 7d, e). Based on different levels of compliance, Participants P2 and C1 reached an average replay performance of 66–81% (P2: $p = 0.001$, one-sided permutation test, 1000 permutations, $\alpha = 0.013$; C1: $p = 0.001$), respectively. Similarly, P2 and C1 reached an average replay performance of 65–76% by grouping sensations according to their temperature level (P2: $p = 0.001$; C1: $p = 0.001$). Although the LDA classifiers did not reach a significant performance for different levels of friction or texture (smooth: cat, apply, key; rough: towel, toast), the replay performances of P2 and C1 did, with average performances of 58%–65%, respectively (P2: $p = 0.010$; C1: $p = 0.001$). Lastly, participant C1 reached an average replay performance of 63% (P2: $p = 0.025$; C1: $p = 0.001$) by grouping sensations based on their macro structure (round: cat, apple, towel; edged: key, toast). These results suggest that participants attempted to design intuitive sensations, as the similarities between stimulus profiles reflected the expected tactile similarities of their target objects.

We wondered whether the visual presence of the object affected the verbal reports of the participants. To quantify this difference, the participants with spinal cord injury filled out the same survey (Fig. S10) as the participants with intact somatosensation in response to their created sensations with and without an image that they thought best matched the evoked sensation. We found mixed effects; only participant P3 significantly changed their ratings in the presence of visual information ($p = 0.011$, two-sample Kolmogorov–Smirnov test, $\alpha = 0.025$, $D = 0.40$, Fig. 8a), whereas P2 ($p = 1.000$, $D = 0.07$) and C1 ($p = 0.760$, $D = 0.17$) did not. However, both P3 ($p = 0.018$, $D = 0.49$, Fig. 8b) and C1 ($p = 0.007^{-1}$, $D = 0.60$, Fig. 8b) had more reliable survey responses when a visual context was provided. This was not true for participant P2 ($p = 0.285$, $D = 0.47$). These results suggest that when the stimulation-evoked sensations were not very distinct—P3 was no better than chance in the replay task and had large variations in selected stimulus parameters—the presence of a visual context skewed sensation reports towards the expected tactile characteristics based on the presented object image. In all other cases, as shown by the robustness of P2 and C1's survey responses to a visual context and their significant replay performances, microstimulation of the somatosensory cortex alone can evoke distinct object-appropriate tactile perceptions.

### Discussion

In this study, people with microelectrode arrays implanted into area 1 of the somatosensory cortex were able to self-select complex microstimulation parameters to create sensations that mimicked tactile properties of natural objects. This was accomplished using a robust experimental paradigm that required participants to search through

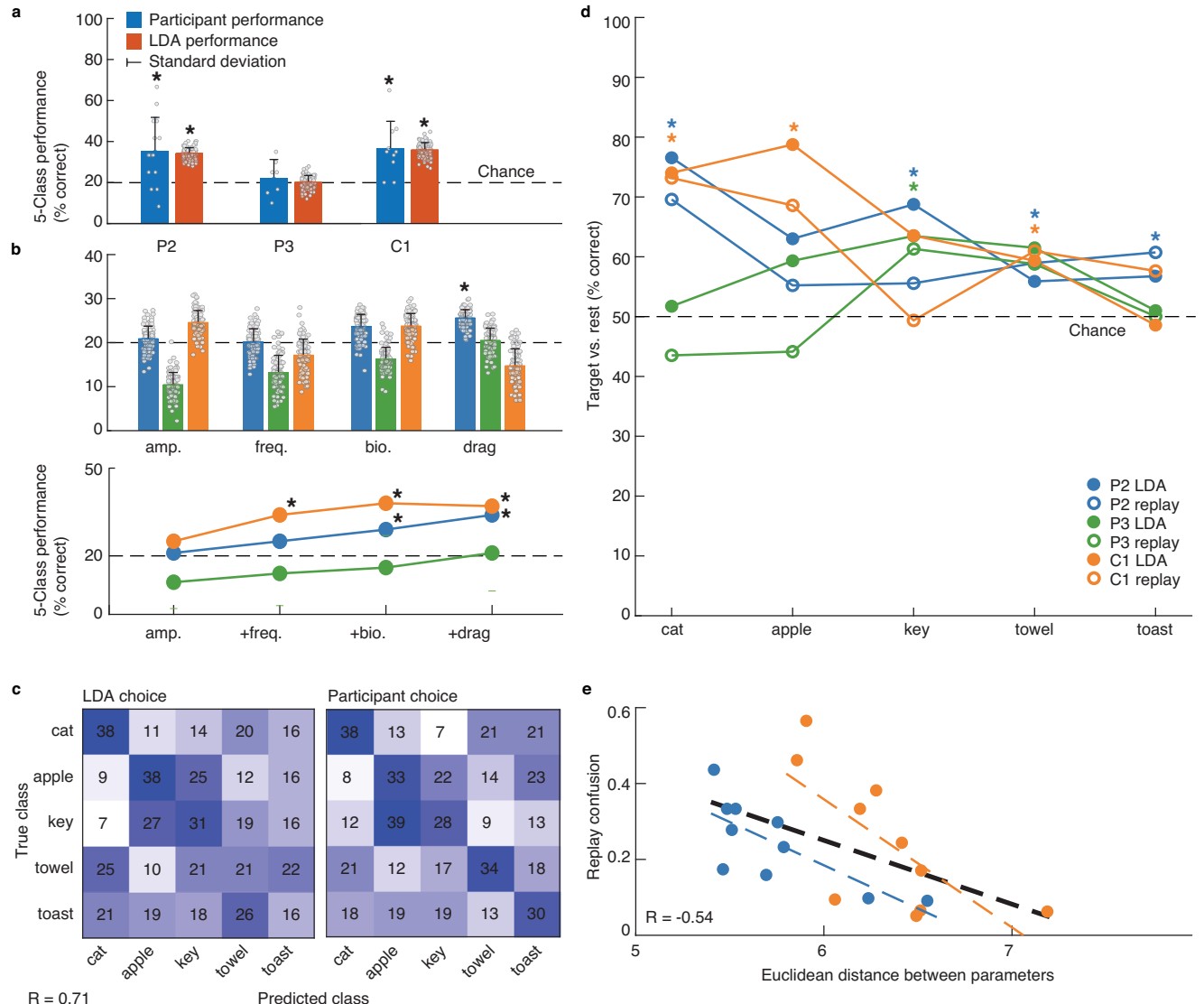

**Fig. 4 | Linear Discrimination Analysis (LDA) and participant replay performance.** Source data are provided as a Source Data file. **a** Mean performances (bar) on identifying which object was represented by a customized sensation. In (**a**, **b**), vertical lines indicate the standard deviation across 100 bootstrapped LDA folds or replay sessions. The LDA performance of both P2 ($p = 0.002$, one-sided permutation test with 1000 permutations, $\alpha = 0.05$) and C1 ($p = 0.002$) was significant (*). **b** Mean LDA performance (bar) on individual (top) or different combinations (bottom) of stimulus parameters, for each participant (color). The combinations include: amplitude; amplitude and frequency; amplitude, frequency and biomimetic factor; or all parameters. Significant (*) performances were found for P2's drag parameter ($p = 0.005$, one-sided permutation test with 1000 permutations without replacement, Bonferroni corrected at $\alpha = 0.007$, LDA accuracy = 25%), C1's combination of amplitude and frequency ($p = 0.001$, LDA accuracy = 34%), and P2 ($p = 0.004$, LDA accuracy = 29%) and C1's ($p = 0.002$, LDA accuracy = 38%)

combination of amplitude, frequency and drag. **c** Confusion matrix for the LDA and participant performances, showing the mean percentage of correct choices across all participants. A positive correlation was observed between the replay and LDA performances (Pearson's r(23) = 0.71, $p = 0.001^{-1}$). **d** Participant and LDA performances across 100 bootstrapped sessions on identifying a target object sensation from all others. Participant P2 performed significantly (*) on cat ($p = 0.001$, one-sided permutation test with 1000 permutations, Bonferroni corrected at $\alpha = 0.01$, 77% LDA accuracy; $p = 0.001$, 70% replay accuracy), key ($p = 0.008$, 69% LDA), towel ($p = 0.005$, 59% replay) and toast ($p = 0.010$, 61% replay) sensations. C1 on cat ($p = 0.002$, 74% LDA; $p = 0.001$, 73% replay), apple ($p = 0.001$, 79% LDA; $p = 0.001$, 69% replay), and towel ($p = 0.002$, 61% replay) sensations, and P3 only on key-sensations ($p = 0.006$, 61% replay). **e** Two sensations are easily confused during replay when their stimulus parameters are more similar, as measured by their Euclidean distance (Pearson's r(20) = −0.54, $p = 0.014$).

four stimulation parameters in a way that could not be learned, and that also provided a meaningful visual context for the five virtual objects (Fig. 1). Participants P2 and C1 selected distinct stimulation profiles per object (Fig. 3) that evoked recognizable and distinct touch percepts in the absence of a visual context (Fig. 4). Like P2 and C1, participant P3 was very satisfied with his created sensations (Fig. 6a). However, for most sensations he was unable to correctly recognize what object they belonged to in the replay task (Fig. 4a, d). His chance-level performance can be explained by insufficient exploration in the object-sensation mapping task (Fig. 2b–d). If a participant does not

explore the full parameter space on each trial, they cannot make an informed choice. The absence of a significant relation between the chosen stimulation settings and objects, and the participant's inability to recognize what object a previously created sensation belonged to shows that his quick search strategy was not effective. In contrast, the object-specific stimulus selections of P2 and C1 show that our method can be effective, provided that the participant explores the full extent of the parameter space.

One could argue that participants simply tried to create distinct sensations – rather than intuitive or naturalistic ones−so that they

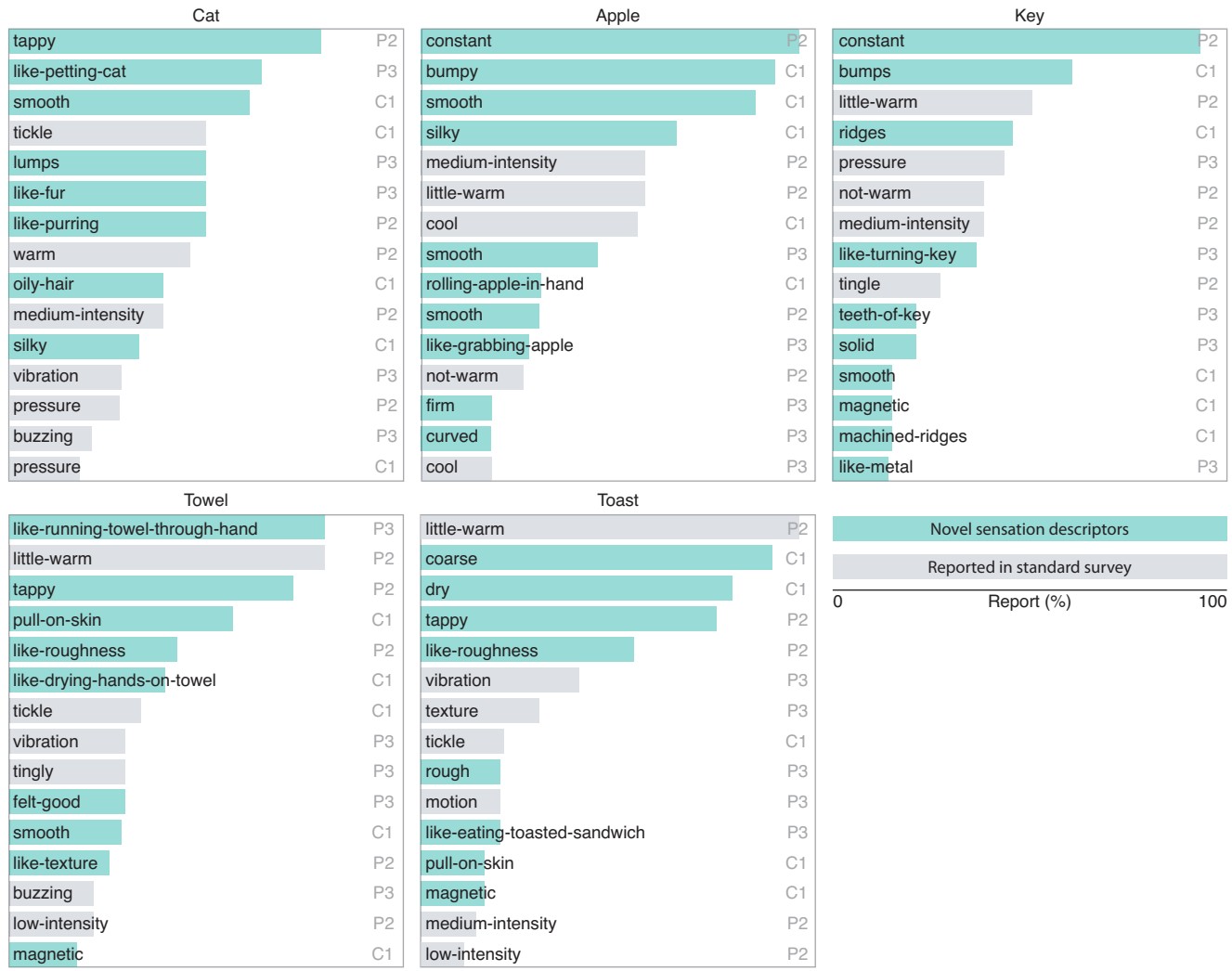

**Fig. 5 | Verbal descriptors of the artificial sensations created using the interface.** The top five verbal descriptors of each participant are displayed for each object. Each bar indicates the percentage of times participants used these descriptors in their report of a sensation for a specific object. Descriptors that were different compared to those reported in previous intracortical microstimulation studies that used standard stimulus trains that were devoid of a meaningful visual context[8,10,22] are highlighted. Source data are provided as a Source Data file.

could successfully perform the replay task. It would have been relatively easy for the participants to do this; the just-noticeable differences in amplitude and frequency are about $10-20\,\mu A$[24,25] and $3-60\,Hz$[12], respectively, and the amplitude and frequency steps they could control were $8\,\mu A$ and $14\,Hz$, respectively. Creating five distinct sensations by manipulating one or two stimulation parameters should have been trivial. However, when the participants were asked to create sensations that represented each displayed object, the stimulation parameters could predict both compliance and temperature across the object set, reflecting the innate characteristics of these objects (Fig. 7d, e). This does not mean that the created sensations necessarily felt like something cold, warm, soft or hard. However, in combination with the congruent vivid sensation reports (Fig. 5), these results do suggest that the created sensations contained some intuitive tactile resemblance to their target object. Further, objects with more similar tactile characteristics were more often confused with each other during the replay task (Fig. 7a). These lines of evidence strongly suggest that the participants both attempted to and were successful at creating intuitive sensations; they did not seek to simply create distinct sensations.

The stimulation parameters that the participants chose overlapped between different objects (Fig. 3). In contrast, the subjective tactile experiences of these objects, when paired with vision, were distinct and vivid (Fig. 5). This suggests that with the right context, ICMS-induced sensations can give rise to realistic touch experiences. Compared to previous research, this experiment was different in two key ways: participants were in control of their own stimulation and participants could actively explore an object that was presented visually. During passive stimulation (no vision, no exploration), people typically report skin-level sensations, such as "pressure" or "vibration", as their attention is focused on their body[8,10]. During active exploration however, their attention is likely focused on the external world, thereby interpreting these same percepts as object-oriented sensations, such as compliance and roughness. The difference in experimental context could explain why the participants in this study spontaneously reported more object-oriented sensation descriptors (Fig. 5). If you are asked to create a sensation of a cat and you touch a picture of a cat while you experience that sensation, you are likely to say it feels "soft", "warm" and "hairy". However, we believe that this is not much different from a person with intact somatosensation who is petting a cat in a real-life situation. In that case, they also see and touch a cat. Through repeated association throughout their life, they have come to experience or describe that particular multi-sensory experience as feeling "soft", "warm" and "hairy". The perceptions that are

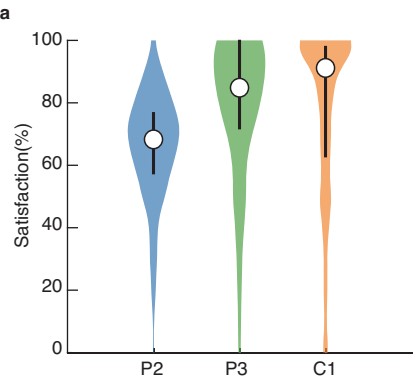
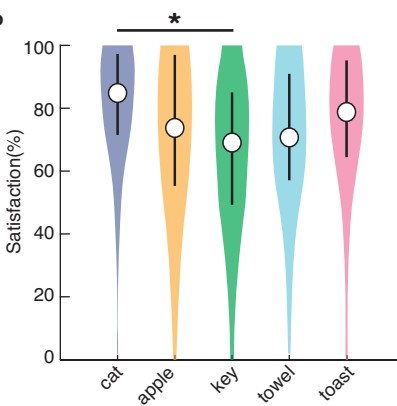
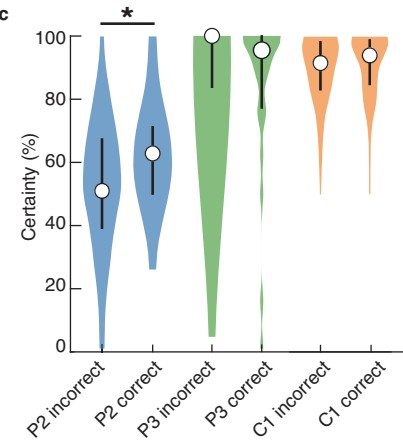

**Fig. 6 | Normalized satisfaction and certainty ratings.** In each figure panel, the 25–75% quartiles of the data are displayed as a thick line around the median. Source data are provided as a Source Data file. **a** Distribution of normalized satisfaction ratings for each participant (P2: 152 trials; P3: 95 trials; C1: 104 trials). The median satisfaction score for each participant is displayed as an open circle. **b** Normalized satisfaction ratings per object across all participants (71 cat, 71 apple, 70 key, 67 towel and 72 toast sensations). On average, participants were more satisfied with the sensations for a cat than for a key ($p = 0.005^{-1}$, two-sided Kolmogorov–Smirnov

test Bonferroni corrected at $\alpha = 0.005$, $D = 0.33$). No other differences were significant. **c** Self-rated certainty with which a participant could (correctly or incorrectly) identify the object corresponding to a set of stimulus parameters in the replay task. Participant P2 was significantly more certain of his answers when they were in fact correct, compared to when they were not ($p = 0.003$, Wilcoxon rank sum, $\alpha = 0.025$, $z = 2.73$, 64 correct and 116 incorrect trials). No significant differences were found for P3 (25 correct and 81 incorrect trials) or C1 (71 correct and 10 incorrect trials).

artificially generated and described in our experiment may not be that different. At the very least, our participants seemed convinced of the appropriateness of their created sensations within the given experimental context (Figs. 5 and 6a). As people will ultimately use closed-loop brain-computer interfaces in daily life, it is important to investigate the functional use and experience of ICMS-evoked sensations in their relevant multi-sensory context.

The sample of available participants for this type of research is limited. Across the globe, there were about seven participants with bi-directional intracortical implants in their somatosensory and motor cortex available at the time of this study. Three of these participants were included in this study, which is a uniquely large sample compared to previous research[8–10,16,26]. Extending the sample beyond this number was simply not possible at the time. Given the observed discrepancies between participants, testing not only a larger participant sample but also a more elaborate set electrodes per participant would be highly desired. For example, the chance-level replay performance of P3 could also be due to the particular set of electrodes used in this study. Identical stimulation parameters can evoke a wide variety of different sensory qualities depending on the electrode location[8,10,16]. Therefore, the choice of electrodes may influence the effectiveness of our method. The set of electrodes of P2 and C1 may have evoked a wider or more distinctive range of sensory qualities compared to those of P3.

Participants explored a large parameter space in a blinded fashion. Because the parameter space was randomized on each trial, participants could not learn a mapping between their movements in the space and their perceived sensory characteristics. Although this manipulation was crucial to ensure that participants made their choices based on tactile perception, rather than vision, it also made the experimental task very challenging and may have contributed to the large variance in the stimulation parameters that represented particular objects. Although the object-specific stimulus trains were recognizable within a single session, participants could recognize only a few stimulus trains correctly that were selected from multiple previous sessions. As such, the significantly larger variation in stimulus parameters across compared to within sessions seemed to complicate replay performance (Fig. S7). However, only a limited number of stimulus trains were tested in these across-session replay tests and additional research is needed to better determine how the sensation

quality may change across days. One way to strengthen the replay task would be to include catch trials; trials that present a random stimulus profile. The replay performance on these catch trials would serve as a baseline to compare the other sensations against; whereas we do not expect randomly created sensations to show a consistent link to a particular object, participant-selected sensations should correlate with particular object choices.

It is possible that our participants found local minima in their exploration of the parameter space, which could explain the high variability within object-specific parameter selections (Fig. 3a). However, the parameter selections of participants P2 and C1 were significantly distinct per object (Fig. 3), showing a valid relation between the selected stimulus parameters and target objects across sessions. This was confirmed by the significant LDA performance, showing a higher variability in stimulation parameters across objects compared to within objects (Fig. 4a, d).

The discrepancy between the participants' certainty (Fig. 6c) and replay performance (Fig. 4) may be due in part to the strictness of the replay task. It may be the case that participants were correct about some perceptual qualities and wrong about others. Participants often confused sensations for more similar objects (Fig. 7a) and performed better at recognizing the compliance and temperature differences between objects than recognizing the object itself (Fig. 7d, e). Rather than getting some recognition for the correctly recognized qualities in the replay task, their object selection is simply counted as incorrect. Other forms of evaluation may be better suited to assess the experienced quality of the created sensations. For example: rating the similarity between the artificial sensation and an actual sensation on a sensate area of the hand or grouping sensations with similar qualities together irrespective of their (potentially incongruent) visual representation.

Brodmann's area 1 is responsible for processing tactile input from mechanoreceptors that give rise to percepts like texture[11,27–29]. By activating this region of the brain using electrical stimulation, we have the ability to evoke intuitive sensations that may resemble features of naturally evoked touch[22]. This study shows that stimulation can evoke intuitive percepts that represent a variety of tactile object properties and that visual input contributes to the overall experience of artificial touch. In the future, self-guided stimulation approach may be used to

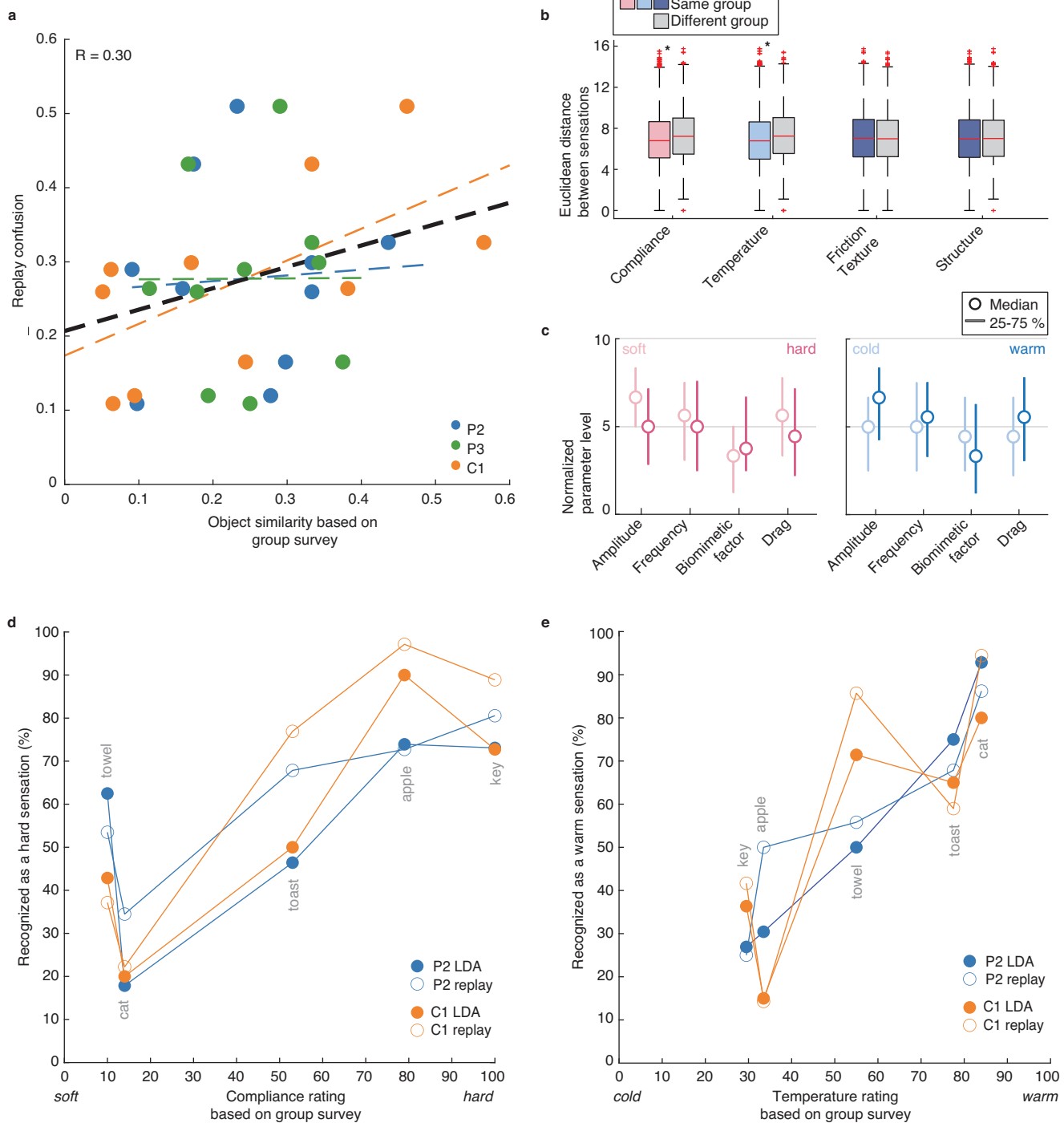

**Fig. 7 | Expected versus predicted object characteristics.** Source data are provided as a Source Data file. **a** For each combination of objects, we compared how often these objects were confused in the replay task (y-axis) with the tactile similarity of these objects (x-axis, where 0 is maximally different and 1 is identical, as determined by the maximum Euclidean distance between two objects based on the mean object quality survey ratings). As object similarity increased, the confusion between sensations increased. **b** Euclidean distance between chosen stimulation parameters for objects with similar (same group) and different (different group) tactile characteristics. The median (central mark), 25th and 75th percentiles (box), data range points (whiskers), and outliers (+ marker) are displayed. Across all participants, the chosen stimulation parameters for objects with similar compliance or temperature were significantly more similar to each other than those of objects with different compliance ($p = 0.001^{-13}$, one-sided Wilcoxon signed rank test, Bonferroni corrected at $\alpha = 0.013$, $z = -8.181$, 8865 same and 5031 different samples) or temperature ($p = 0.003^{-21}$, $z = -10.104$, 9306 same and 4917 different

samples). No differences were found in friction/texture (8693 same and 5071 different samples) or structure (6989 same and 5050 different samples). **c** Median normalized stimulation settings across all participants for different levels of compliance and temperature. The thick lines indicate the 25–75 percentiles of the data. **d** For each object, we compare the median compliance rating from the object quality survey (x-axis) with the percentage of object sensations recognized as belonging to a group with certain tactile characteristics (y-axis). The filled circles indicate the percentage of stimulus parameter sets for each object that were ascribed a certain compliance (145 hard vs. 100 soft sensations) by the LDA classifier for participant P2 and C1. The open circles show how P2 and C1's choices in the replay task matched the compliance predictions of the object quality survey. Participant P3 is excluded from this follow-up analysis due to their chance level performance on the replay task. **e** Same as (**d**), but for object temperature (93 cold vs. 152 warm sensations).

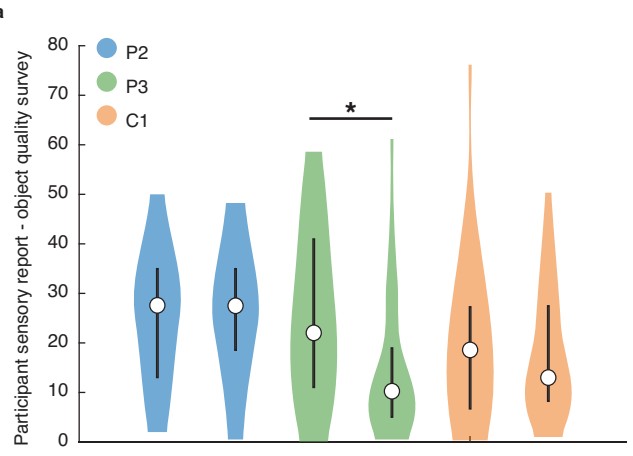

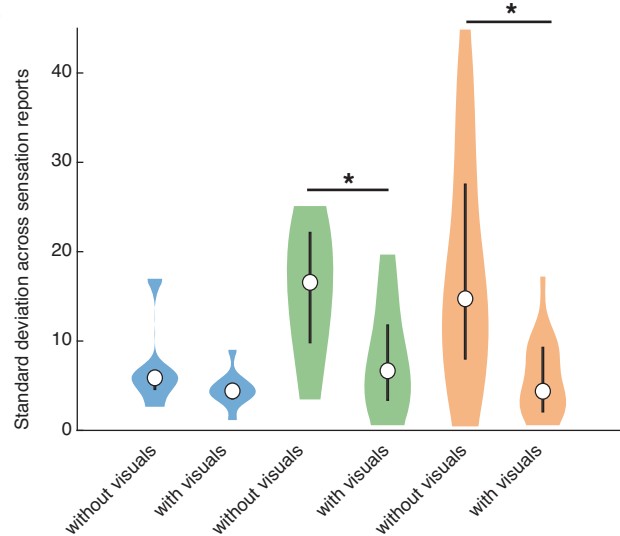

**Fig. 8 | Effect of visual context on perceived tactile characteristics.** Source data are provided as a Source Data file. **a** Distribution of mean absolute differences between the survey ratings of participants P2, P3, and C1 with (30 trials per participant) and without (30 trials per participant) a visual context, and the survey ratings of the participants with intact somatosensation, across all objects. The survey ratings with and without a visual context were significantly different for participant P3 ($p = 0.011$, two-sample Kolmogorov–Smirnov test, $\alpha = 0.025$, $D = 0.40$). **b** Distribution of mean standard deviations across all survey ratings of participant P2 (6 samples without and 10 samples with visual context), P3 (18 samples without and 17 samples with visual context) and C1 (23 samples without and 18 samples with visual context) with and without an image that they thought best matched the evoked sensation. Median values are displayed as an open black circle and the 25–75% quartiles are displayed as a thick black line. Both P3 ($p = 0.018$, two-sample Kolmogorov–Smirnov test, $\alpha = 0.025$, $D = 0.49$) and C1 ($p = 0.007^{-1}$, $D = 0.60$) were significantly more consistent in their survey ratings with a visual context, compared to without.

efficiently characterize percepts evoked by novel stimulation paradigms.

## Methods

### Participants

This study was part of a multi-site clinical trial, registered at Clinical-Trials.gov (NCT01894802). The purpose of this trial is to collect preliminary safety information and demonstrate that intracortical electrode arrays can be used by people with tetraplegia to both control external devices and generate tactile percepts from the paralyzed limbs. This manuscript presents the analysis of data that were collected during the participants involvement in the trial but does not report clinical trial outcomes. The study was conducted under an Investigational Device Exemption from the U.S. Food and Drug Administration and ethically approved by the Institutional Review Boards at the University of Pittsburgh and the University of Chicago. Ten years prior to the time of implantation, participant P2 sustained a C5 motor/C6 sensory ASIA B spinal cord injury. He was between 25 and 30 years old at the time of implant and between 30 and 35 years old during the data collection for this study. Participant P3 was of similar age as P2 during implant and data collection and sustained a C6 ASIA B spinal cord injury 12 years prior to implantation. Participant C1 sustained a C4 ASIA D spinal cord injury 35 year prior to implantation. He was 55–60 years old at the time of implant and during data collection. All participants had some residual sensations in their right hand, although they described these sensations as feeling non-natural. Prior to any study procedures, all three participants provided their informed consent.

### Stimulation apparatus

Prior to this study, participants were implanted with two 2.4 × 4 mm microelectrode arrays (Blackrock Microsystems, Salt Lake City, UT) in Brodmann area 1 in their left hemisphere[10,24]. Another two arrays were implanted in their motor cortex within the same hemisphere, but were not used in this study. Each sensory array contained 60 1.5 mm long electrode shanks, wired in a checkerboard pattern across a 6 by 10 grid

such that 32 electrodes could be stimulated. The electrode tips were coated with a sputtered iridium oxide film. The stimulation return electrode was a titanium pedestal that was fixed to the skull. Upon stimulation, participants reported sensations on their right hand. The quality and location of these sensations depends on the stimulation protocol[9,10,19]. The stability of these sensations is measured each month using a survey. To do so, each stimulation electrode is stimulated one-by-one with a standard stimulus (60 μA, 100 Hz, 1 s), after which the participant reports the experienced location and quality of the resulting sensation (if any). Based on these historical data, a single set of three electrodes was selected per participant for the current study. These electrodes were chosen to maximize the reliability and intensity of the evoked sensation, meaning that a clearly detectable sensation was evoked (almost) every time the electrode was stimulated. Furthermore, these electrodes were chosen to evoke sensations across the palmer side of the right hand (Fig. 1b).

A CereStim C96 multi-electrode microstimulation system (Blackrock Microsystems, Salt Lake City, UT) was used to deliver stimulation. Each stimulation pulse was current-controlled and charge-balanced, consisting of a 200 μs cathodal phase followed by a 100 μs interphase period and a 400 μs anodal phase set to half of the amplitude of the cathodal phase. Electrodes could be stimulated up to 275 Hz at amplitudes ranging from 2 to 100 μA[10,25,30].

Neural data were collected using NeuroPort Central software (Blackrock Microsystems, Inc.). These data, along with behavioral data, were formatted and saved for analysis using our own code toolbox in Python 3.7.6 and Matlab 2021a.

### Stimulation protocol

During this study, participants were able to manipulate four ICMS parameters in real-time: amplitude, frequency, biomimetic factor and drag (Fig. 1c). Each of these parameters could be manipulated across 10 levels. The minimum amplitude was set to 10 μA for all participants, which is just below or at detection threshold for most electrodes[19]. Predetermined individual safety limits set the maximum amplitude to 100 μA for P2, 80 μA for P3 and 90 μA for C1.

To accommodate for potential differences in the perceived intensity of a stimulus at the same stimulus current across electrodes, participants were asked to equalize their perceptual intensity across the three electrodes at the start of each session. One electrode would serve as a base with a fixed amplitude of 48 µA for P2 and 44 µA for P3 and C1. Participants were asked to match the perceived intensity of the other two electrodes to the base electrode using two sliders that manipulated their respective amplitudes. The participants could stimulate each electrode for 1 s whenever they wanted to. Participants were not told that they were manipulating the amplitude level of different electrodes. Moreover, the electrode assignment to stimulation buttons 1, 2 or 3 was randomized so participants were not aware what electrode they were manipulating. Each electrode served as base electrode once during a block of three trials. Based on the resulting amplitude ratio, an individual amplitude range was determined for each electrode. For example, with an amplitude ratio of 48(base):40:60 = 0:−8: + 12 µA, the ranges for each electrode would be: electrode 1 = [18, 88] µA, electrode 2 = [10, 80] µA, and electrode 3 = [30, 100] µA. The final amplitude ranges were set to the mean across the amplitude ranges obtained at each of the three trials. For each electrode, 10 values were chosen that equally divided the determined amplitude range, e.g., electrode 1 = [18, 26, 34, 42, 50, 56, 64, 72, 80, 88] µA. To control for possible differences in the perceived intensity that may occur over the course of a single session, the task was repeated at the end of the session. No significant differences were found between the task results at the beginning and end of a session (Fig. S11a).

The stimulation frequency was divided into ten increments between 20 and 150 Hz and the same frequency was used on all three electrodes in all trials. Because frequency may have electrode-dependent effects on perception[19], we delivered single-electrode stimulus trains at 20, 85 or 150 Hz and asked participants to indicate on a screen which stimulus was the most intense. This was done for all three electrodes in a random order at an amplitude of 62 µA and this task was repeated at the end of the session. Although participant C1 consistently rated a high frequency (150 Hz) stimulus as most intense on all electrodes, the results of participants P2 and P3 were mixed. Participant P2 seemed to have a low frequency (20 Hz)-preferring electrode, a high frequency (150 Hz)-preferring one and a medium frequency (85 Hz)-preferring one. Participant P3 seemed to have two high frequency (150 Hz)-preferring electrodes and one medium frequency (85 Hz)-preferring one. Except for electrode 3 of participant P2, there was no significant difference between the task results at the start and end of a session, suggesting that the frequency and electrode specific differences in intensity were consistent over time (Fig. S11b).

The drag parameter determined how long an electrode remained active after the cursor left its receptive field on the digital object. The total time that an electrode was stimulated was calculated as: $E = T + D*T$, where $T$ was the total time that the cursor was on the receptive field of that electrode (Fig. 1b), and $D$ was a participant-selected drag factor between 0 and 2. By increasing drag, multiple electrodes could be active at the same time.

Neural activity in the somatosensory cortex typically contains a clear onset and offset transient in response to a mechanical indentation of the skin[15]. The magnitude of the onset transient is about 15 times as big as that during the sustained period of skin contact, whereas the offset transient is about 8 times as big. We mimicked these naturally occurring patterns of neural firing activity via a biomimetic factor that specified the amplitude difference between the onset and offset transients and the amplitude during sustained object contact. The new amplitude was calculated as $Ab = k1*(\frac{dP}{dt}) + k2*(P*A)$, where $k1$ was defined as $k1 = \frac{A}{V}*B$ for the onset and $k1 = \frac{\frac{8}{15}*A}{V}*B$ for the offset transient, $B$ was the participant-selected biomimetic factor between 0 and 10, $dP$ was the change in pixel color as the cursor moved across an object mask (Fig. S12), $dt$ a fixed time interval of 20 ms, $k2$ determined

the maximum amplitude during sustained touch as $k2 = B*(\frac{1}{15}*A)$, $P$ the current masked pixel color (Fig. S12), $A$ the participant-selected amplitude, and $V$ was a constant of 5 that indicated the maximum expected velocity. For each object, a black and white mask was created, defining an onset and offset gradient that was appropriate for its corresponding compliance level (Fig. S12). The pixel values were taken from this mask image. When the biomimetic factor was at its maximum, it was maximally mimicking the biologically inspired amplitude where the onset transient was set at a maximum of 15× the amplitude during sustained touch and the offset transient is at maximum 8× the amplitude during sustained touch. In case of an unexpected drastic velocity increase, the amplitude could never exceed the participant-selected amplitude level. The amplitude during sustained touch was set to 1/15× the participant-selected amplitude.

## Experimental procedure

Our experimental design is inspired on Shokur et al.[21], who asked participants with paralysis of both legs due to spinal cord injury to create sensations for walking on grass, pavement or sand using vibro-tactile stimulation on their lower arms. Similar to Shokur et al., four different ICMS parameters were mapped to the x- and y-axes of two rectangles. Due to the limited movement capabilities of our participants, the rectangles were presented on a Windows Surface laptop (Fig. 1a). By moving around a cursor in each rectangle, participants could adjust the stimulation parameters in real time. Above the rectangles, one of five objects was displayed: apple, toast, towel, cat, or key (Fig. 1a, d). Participants were asked to find the best possible stimulation settings that represented these objects while they interacted with it; we called this the "object-sensation mapping task." Participants could trigger ICMS by touching the virtual object using a stylus, or by using an automatic stimulation option that caused a digital cursor to move back and forth across the object every 6.5 s, roughly evoking stimulation for 4 s with a 2.5 s break (Fig. 1a). Hidden from the participants view, the presented object was divided into three regions: as long as the cursor was within one field, a specific electrode would be stimulated (Fig. 1b). The participants received stimulation during only a subset of the total trial time; whenever they interacted with the presented object via their own hand or the automated cursor.

For each participant, we selected three electrodes that evoked sensations in the fingers and or palm of the right hand[10,19]. Moving a cursor across the digitally presented object created a sensation across the palmer side of the right hand (Fig. 1b). Allowing participants to make their own explorative movements across the object was done to create a more realistic and interactive experimental setting. Depending on how participants moved the cursor across the presented object, they could influence the location, biomimetic factor, drag and overall length of stimulation. As our participants still had some control of their arms, they performed this task using the back of their hand or a stylus. To prevent actual sensory input from interfering with the ICMS-evoked sensations, participants controlled the interface using their left hand.

At the start of each object-sensation mapping trial, the parameter assignment to the rectangle axes was randomized. In addition, each parameter distribution was randomly flipped (i.e., running from minimum to maximum, or from maximum to minimum on a particular rectangle axis). This randomization forced the participants to base their decisions on their sensation alone, rather than some memorized visual position of the cursors within the rectangles. At the end of each trial, after participants finalized a stimulation setting for an object, they were asked to rate how satisfied they were with their created sensation using a slider that ran from "unsatisfied" to "satisfied" (Fig. 1c).

Each object was repeated two to three times per session in the object-sensation mapping task. Participant P2 typically created three sensations for three different objects per session. Each session, three out of five objects were randomly selected. P2 thus completed nine object-sensation mapping trials per session. In contrast, participants

P3 and C1 created two sensations for all five objects per session, summing to ten object-sensation trials per session. However, people with tetraplegia deal with frequent and unpredictable health issues. For these reasons, scheduled test sessions sometimes needed to be adjusted on the spot. Depending on the participants' physical and mental state, test sessions could sometimes run longer or had to be shortened. Across 22 sessions, participant P2 created 30 cat, key and toast sensations, 33 apple sensations and 29 towel sensations (Fig. S2). Across 10 sessions, participant P3 created 20 cat and toast sensations, 18 apple and towel sensations, and 19 key sensations. Similarly, participant C1 created 21 cat, 19 apple, key and towel sensations and 20 toast sensations across 10 sessions (Fig. S2).

To test whether the sensations were unique and distinguishable for the different objects, the stimulus parameters sets for each object in a session (regardless of their satisfaction score) were replayed to the participants at the end of that session. In this 'replay task', the participants experienced the ICMS-evoked sensations without any corresponding visual stimulus; a grey rectangle was shown instead of the original object (Fig. 1d). Participants were then asked to indicate what object best matched the experienced sensation. After each object selection, they were asked to indicate how certain they were of their choice, using a slider that ran from "uncertain" to "certain". Each sensation that was created earlier during that session in the object-sensation mapping task, was tested at least twice in the replay task. Therefore, the replay task typically consisted of 18 trials for P2, and 20 trials for P3 and C1 (Fig. S2b).

To investigate the stability of the sensations across time, as well as their dependence on visual context, participants completed two additional sessions of the replay task three (P3, C1) to seventeen (P2) weeks after finishing data collection. During these sessions, participants were presented with three parameter sets per object: the sensation with the highest satisfaction rating in the first and second halves of the experimental sessions, and the sensation that was closest to the median of the selected parameters for that object. First, participants were asked to rate (Fig. S10) the experienced sensations using the same tactile dimensions as used in the online survey with control participants (Fig. S1a). Then, participants had a chance to familiarize themselves with the previously created sensations and go through each of them at their own pace while being presented with their correct visual context. Lastly, participants performed a modified version of the replay task in which they could select and explore the object image that they thought best matched the evoked sensation. After assigning an object to a sensation, thus providing their own visual context, participants were asked to fill out the same survey as before (Fig. S10). This replay task was repeated another two times without the survey, providing a total of three repetitions per unique sensation. Finally, during one of the sessions, all participants completed a short survey on their experience during the experiment (Fig. S10).

## Statistical analysis

All data were analyzed using our own code in Matlab 2020b. A two-sided Kolmogorov–Smirnov test was used to test whether the collected pre- and post-session amplitude ranges differed significantly per participant. The significance level was set to 0.025, as a single comparison was made per participant. A similar strategy was used to evaluate the pre- and post-session frequency intensity choices per electrode. However, as these differences were assessed for each of the three stimulation electrodes per participant, the significance level was Bonferroni-corrected[31] and set to 0.025/3 = 0.008.

We compared the trial completion times, aggregated stimulation times per trial and total percentage of explored parameter space across participants using two-sided Wilcoxon rank sum tests. A single test was performed for each of these three measured variables. After Bonferroni correction, the significance level was set at 0.025/3 = 0.008.

To assess differences between stimulation parameters that were chosen during the object-sensation mapping task, we used a two-sided Kruskal–Wallis test to assess whether the individual parameter selections for different objects came from the same or a different distribution. Only trials that had a normalized satisfaction score of at least 50 out of 100 were included in our analysis. To compensate for multiple comparisons across each of the four stimulation parameters, the Bonferroni corrected significance level was set to 0.025/4 = 0.006. If a significant difference was found, post-hoc one-sided two-sample Kolmogorov–Smirnov tests were used to determine what object-specific parameter selections were different from each other, testing each possible combination of two objects: cat-apple, cat-key, cat-towel, cat-toast, apple-key, apple-towel, apple-toast, key-towel, key-toast, towel-toast. The Bonferroni-corrected statistical significance level was set to 0.05/10 = 0.005.

To investigate the separability of object-specific stimulus profiles, we calculated the Euclidean distance between different groups of parameter selections. First, we selected only those stimulus profiles that were recognized consistently correctly in the replay task; meaning that the correct object was assigned to a sensation in the replay task for at least two out of three repeated presentations. Then, the Euclidean distance between each possible combination of stimulus profiles was calculated within each object category (e.g., cat-cat, apple-apple) and across different object categories (e.g., cat-apple, towel-toast). For each participant, a one-sided nonparametric Wilcoxon rank sum test with a significance level of 0.05 was used to assess whether the Euclidean distance was smaller within an object category compared to across different object categories. In addition, we calculated the Euclidean distance between each sensation and the median across all successfully identified stimulus profiles of its target object. Again, we used a one-sided nonparametric Wilcoxon rank sum test with a significance level of 0.05 to assess whether the distance between individual sensations and the median target stimulus profile was smaller for correctly compared to incorrectly recognized sensations in the replay task.

To determine whether a specific object could be predicted from a stimulation parameter set, a linear discriminant analysis (LDA) classifier was trained and tested on the object-specific parameter selections of each participant in two steps. In step one, tenfold cross validation was performed on the bootstrapped data with their original class labels. Because the sample size in this study was small, collecting only a maximum of 30 samples per object, the cross-validation procedure was repeated 100 times and the results were averaged to get a stable approximation of the true classifier performance. At the start of each validation run, the class samples were supplemented to match the maximum class size with randomly drawn samples with replacement from the same object distribution. Step two conducted a permutation test with 1000 permutations. For each permutation, a random sample of the maximum class size was drawn without replacement from all stimulation-parameter selections, thus randomly assigning stimulus parameters to object classes. Then, tenfold cross validation was performed on this sample, generating a null distribution of possible LDA classification performances. The original LDA performance calculated in step one was considered significant if it was greater than 95% of the LDA performances on the permuted samples of step two[32] To check for the individual contributions of each stimulation parameter to the significant LDA performances, an LDA classifier was trained and tested on all possible combinations of stimulation parameters using the same procedure. Since there was a total of 15 possible combinations of stimulation parameters (amplitude, frequency, biomimetic factor, drag, amplitude + frequency, amplitude + drag, amplitude + biomimetic factor … amplitude + frequency + biomimetic factor + drag), the significance level was Bonferroni corrected to 0.05/15 = 0.003.

To check whether any differences in stimulation parameter selections could be explained by their relative contribution to the total

charge per electrode, differences in total charge per object were calculated based on the replay data. Because participants were free to explore these sensations as they wished by touching a virtual rectangle, there were no fixed-length stimulation trains. To compensate for relative differences in total stimulation time between electrodes, the total charge values were normalized to the total time spent on each electrode. A two-sided Kruskal–Wallis test was used to assess any significant differences in the total charge between different objects. The statistical significance level was set to 0.025. In case a statistical difference was found, multiple post-hoc one-sided two-sample Kolmogorov–Smirnov tests were applied to assess what objects were significantly different from each other. The Bonferroni-corrected statistical significance level was set to $0.05/10 = 0.005$.

The participants' performance on the replay task was assessed by calculating the average percentage of correct answers (the chosen object equals the original object for which the sensation was designed) across all sessions. To assess significance, the participant's performance was compared against a naïve classifier that predicted each possible class with equal probability. Using a permutation test with 1000 permutations, the performance of this naïve classifier was calculated against the underlying true class distribution across all trials. Chance level was established as the mean performance of this classifier across all permutations. The participant's task performance was considered significant if the percentage of correct answers across all trials exceeded 95% of all permuted naïve classifier performances. A one-sided nonparametric Wilcoxon rank sum test was used to assess whether the participants certainty scores were significantly higher in case their answers were correct compared to incorrect[33]. The significance level was set to 0.05, as a single comparison was made per participant. For each participant, we compared the correlation between the perceptual choices of the participant on the replay task and the stimulus-parameter-based choices of the corresponding LDA classifier. To do so, we computed the Pearson's correlation coefficient between the confusion matrices of the participant and the LDA classifier choices.

For each object, we calculated the percentage of object-specific sensations that was wrongfully assigned to the other object for each possible object pair. For example, in case of apple and cat sensations, we calculated the percentage of apple sensations that were wrongfully assigned to a cat object plus the percentage of cat sensations that were wrongfully assigned to an apple across all presentations of apple and cat sensations during the regular replay task. We compared the replay confusion score of each possible pair of objects to the average Euclidean distance between their corresponding stimulus profiles. We computed the Pearson's correlation coefficient to test whether the replay confusion decreased as the distance between stimulus parameters increased.

In addition to the participant replay and LDA performances across each of the five objects, we calculated one-vs-rest performances for each object. To do so, we repeated the same procedures as describes above using binary class labels, where each repetition of a target object sensation was labeled as class 1 and the rest of the sensations as class 2. Since this analysis involved five repetitions, one for each object, the significance level of the permutation test was set to $0.05/5 = 0.01$.

The participants' performance on the across session replay task of the final two sessions was assessed in a similar way to the within session replay performance. In addition, we calculated which sensations were correctly matched to their original target in at least two out of three repetitions. To assess the variation of object-specific stimulus parameters within and across sessions, we calculated the standard deviation across all repetitions of an object-specific parameter within and across sessions for each participant. A one-sided Kruskal–Wallis test (with a significance level of 0.05) was used to assess whether the object-specific variation across all parameters and participants was significantly different within and across sessions. Additional post-hoc one-sided two-sample Kolmogorov–Smirnov tests (Bonferroni corrected significance level was set to 0.01) were used to assess significant differences in the variation of each individual parameter within and across sessions.

Satisfaction scores were normalized to the minimum and maximum scores per participant to account for individual differences. A two-sided nonparametric Kruskal–Wallis test was used to assess whether the satisfaction ratings across all participants came from the same or a different distribution per object[34]. If a statistically significant difference was found, individual post-hoc one-sided nonparametric two-sample Kolmogorov–Smirnov tests were used to assess which objects differed significantly from each other in terms of their satisfaction rating[35]. To compensate for multiple comparisons, testing potential differences between all 10 possible combinations of five object sensations, Bonferroni correction was used[31]. After Bonferroni correction, the level of significance was set to $0.05/10 = 0.005$.

The Euclidean distance of the control participants' survey ratings was calculated for each possible pair of objects. These distance measurements were then normalized to the maximum object-pair distance. To get a measurement of object similarity rather than difference, we subtracted these normalized Euclidean distances from 1 for each object pair. This provided a similarity score between 1 (most similar) and 0 (most dissimilar). We then compared the control participants' similarity scores between two objects to the participant's replay confusion between those two objects. To assess whether object-sensations were more easily confused as the corresponding objects shared more tactile characteristics, we computed the Pearson's correlation coefficient.

In addition, we investigated what tactile characteristics could best explain the observed separability between created sensations. To do so, we calculated the Euclidean distance between stimulus profiles corresponding to objects with different levels of compliance, temperature, friction/texture, and structure. Each tactile characteristic was divided into two levels: minimum (e.g., "smooth") and maximum (e.g., "rough"). These levels were determined by the mean ratings on these tactile dimensions being bigger or smaller than 0, as judged by the participants with intact somatosensation (Fig. S1a). For example, according to the control survey, cats, apples and towels were considered round, whereas keys and toast were considered edged. A one-sided nonparametric Wilcoxon rank sum test with a significance level of $0.05/4 = 0.013$ was used to assess whether the Euclidean distance between sensations in the same tactile group (e.g., round-round, edged-edged), was smaller than between different tactile groups (e.g., round-edged). To further assess the separability of stimulus profiles according to different tactile characteristics, rather than objects, we trained and tested four individual binary LDA classifiers on each of the listed tactile characteristics. Each classifier was trained and tested using the same tenfold cross validation procedure and permutation test as described above. For each object, the percentage of classifier predictions of each tactile characteristic was computed.

Lastly, to assess the influence of visual context on the experience of the customized ICMS-evoked sensations, we calculated the absolute difference in survey ratings between our participants and the median of the survey results from people with intact somatosensation for each object. We did this for the sensations that we presented with and without a self-selected visual context in the final two replay sessions of our participants. In addition, we calculated the standard deviation across the survey ratings of our participants for each object-specific sensation. Significant differences in the absolute mean difference between our participants' and the control participants' ratings, and in the standard deviation in survey ratings across repeated presentations of object-specific sensations were assessed using a two-sided two-sample Kolmogorov–Smirnov test. Significance level was set to 0.025 as a single test was performed on each dataset.

## Ethics and inclusion

The conducted research is globally relevant, as it addresses the restoration of sensation in all individuals living with high-level spinal cord injury. This research has included local researchers throughout the research process. The roles and responsibilities of this research were agreed amongst collaborators ahead of time. The research was conducted under an Investigational Device Exemption from the U.S. Food and Drug Administration and ethics approval was obtained from the Institutional Review Boards at the University of Pittsburgh and the University of Chicago. The research posed some health risks to the participants, similar to those observed with the implantation of other chronic devices such as deep brain stimulators and cochlear implants. The most common complication for these types of devices is infection. This study includes a robust risk management and mitigation plan that is approved by both the U.S. Food and Drug Administration and university Institutional Review Boards. We have taken relevant local and regional research to our study into account in our citations.

## Reporting summary

Further information on research design is available in the Nature Portfolio Reporting Summary linked to this article.

## Data availability

The de-identified data that support the findings of this study are available under restricted access for participant privacy. Access can be obtained upon request to the study PI by an investigator who is prepared to securely handle data resulting from human research. The study PI will respond within 1 month after a request is made. The data are shared via the Data Archive BRAIN Initiative (DABI) under project code JZITRHZ6X9WI at https://doi.org/10.18120/5b2x-2w11. Source data are provided with this paper.

## Code availability

Customized code used for analysis is available through Github at https://doi.org/10.5281/zenodo.14722512. The code used for data collection can be made available upon request to the study PI.

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

## Acknowledgements

We would like to thank the magnificent N. Copeland, Mr. Dom, and S. Imbrie for their participation in this research study. Without these wonderful brain-computer interfacing pioneers, this research would not be possible. In addition, we would like to acknowledge Prof. Bensmaia's support of this project; he unexpectedly passed away during this study and his critical thinking and team spirit were crucial to this project's success. This study was supported by the National Institute for Neurological Disorders and Stroke (UH3 NS107714) and the Dutch Research Council (NWO Rubicon: 019.193SG.011, NWO Vidi: VI.Vidi.191.210).

## Author contributions

Under the supervision of B.S. and R.G, C.V. led the experiment design, implementation, data collection and analysis, and wrote the first draft of the research article. V.K. conducted part of the experiment implementation and data collection. C.G. conducted the data collection with participant C1. S.B. and M.B. provided feedback on the data analysis. All authors provided critical review of the text.

## Competing interests

R.G. is on the scientific advisory board of and Neurowired LLC and had a consulting role with Blackrock Neurotech during this project. The remaining authors declare no competing interests.
