## [Transparent Peer Review file · Nature Communications]

Conveying tactile object characteristics through customized intracortical microstimulation of the human somatosensory cortex

Corresponding Author: Dr Robert Gaunt

Version 0:

Reviewer comments:

Reviewer #1

(Remarks to the Author)

In this manuscript Verbaarschot, Gaunt and colleagues demonstrate a novel paradigm for individuals with bi-directional intracortical brain-computer interfaces to explore the stimulation parameter space. This is an important and timely issue as generating robust sensory responses from intra-cortical microstimulation parameters is key in providing usable feedback to improve prosthetic control. The method shows promise, however, more is required to ensure that these results are robust given the reliance on the subjective experience of a small sample.

My biggest concern in this manuscript is the reliance on the subjective experience of a small sample size to demonstrate that this method is effective in improving the sensory experience of individuals using bi-directional intracortical brain-computer interfaces. This issue is exemplified in the fact that subject P3 spends the least time exploring the parameters space, has no significance in the parameter selection fitting, yet has a higher satisfaction than P2, more similar to that of the highest respondent C1. This indicates that relatively arbitrary parameters, identified quickly could be enough for patients to appreciate, which contradicts the sentiment of the manuscript. This confound should be better discussed in the manuscript. I appreciate that there are very few participants at this stage able to perform experiments such as these, however given this, more objective analysis would be expected, or more participants should be included in such a study before interpreting the results.

The methodology proposed for participants to use shows promise, however there are controls that should be implemented to ensure the robustness of the results presented. There is no mention of catch trials anywhere in the manuscript, which are important to show that the evoked sensations are related to the changing stimulus parameters themselves and not evoked purely based on the expectation of a response.

Following from the previous point, although parameter axis are shuffled and blinded it still seems that significant portions of the parameter space are not explored. Was there any effort made to initialize the participants in different parts of the parameter space? This could indicate that participants fall into local minima in which they are able to interpret the sensations in a way they find satisfying. To address this, the authors may consider performing the experiments again, but sequentially 'leaving out' parameters that participants find satisfying without participants knowing. Either participants will always be able to evoke useful precepts meaning that this method can not overcome the power of sensory expectation and does little to usefully constrain the stimulation parameter space, or participants will eventually reach a parameter space where they are unable to find any parameters for useable sensations in which case the methods presented will be demonstrated more robustly.

The authors here attempt to constrain a large parameter space when considering stimulation evoked sensations. However, by adding 5 more objects they in fact increase the parameters space problem 5-fold! Although significant, the classification accuracy of the LDA model seems low and potentially the significance is driven by the low chance level achieved by having multiple classes. The authors may consider a confusion matrix showing of the significance of the LDA performance if only pairs of classes are tested.

In order to utilize bi-directional brain-computer interfaces over time and minimize training time, a lot of effort recently has

been placed on the ability to decode longitudinally and across participants. The same could be true for evoking sensations. The authors discuss a poor longitudinal stability for the stimulation parameters, but I would be interested if the authors looked at training on one participant and testing on the others to see how stable the parameter selection is across participants.

Line 40: 'In the few human studies that exist, participants reported sensations like "buzzing", "tingle", and "pressure" in response to simple stimulation'. I do not believe this statement fairly reflects the broad range of naturalistic sensations captured in references 8,10,16.

The 'cat' label is missing in figure 2b

Reviewer #2

(Remarks to the Author)

Verbaarschot and colleagues present a well-executed and interesting set of experiments and results. Their attempt to address questions related to phenomenology and naturalness in artificial somatosensation restoration via ICMS is timely and exciting.

Comments

Major

If I understand correctly, the user in this protocol used only one hand to define the four parameters. In the current study, it is not clear if they had to switch between two tablets if they fixed the first two parameters and then the last two (or if they switched between the two). Do you think this sequential choice of parameters could affect the results? Would non-sequential (e.g., 3D control + rotation of the left hand - assuming the patient was able to do it) have changed something?

The verbal descriptors reported in Fig 5 are exciting. If I understand correctly, the standard survey is the one the same authors presented in their previous papers published in 2016 and 2021. The broadening of the reported sensations compared to the standard survey is very interesting, but is it possible that it is a simple effect of the presence of the objects that they chose to work with? Do participants 'simply' give new descriptions because they are shown a cat, or is there more to it?

Minor

Line 51: References 8, 10, 17, and 18 seem misplaced. Are the authors saying that all these experiments provided engaging experimental paradigms?

Introduction: Can you please describe the challenges in terms of the 'phenomenology' of the perceived feedback when using ICMS?

Line 108: What is the difference between time exploring and the amount of stimulation?

Could the difference in score be explained by the somatotopic projection of the three stimulated areas? Could that explain the rather poor results of P3 (for whom the blue and orange stimulation points are more overlapping)?

In Fig 2B, the label 'cat' is missing.

Please follow the same order of materials as much as possible throughout the figures.

Curiosity: What justifies the choice of n=34 for the healthy participants group?

Lines 192-198: I did not understand how you get the ground truth here. How do you find the values for the orange and blue circles? Where in the replay or the LDA is the information about hardness and temperature?

Did the participants have residual sensation in their hand (in particular C1)? How did they relate the phenomenological experience in real life versus via ICMS?

Reviewer #3

(Remarks to the Author)

The authors present a study on sensory stimulation of participants with tetraplegia by means of invasive brain electrodes. The main novelty of the study is that participants were able to choose the stimulation parameters for different objects after an exploration phase.

The authors support their results with many tests and motivate their findings with different analyses. However, while the manuscript is overall well written and enjoyable to read, it is sometimes hard to get the reasons behind some methodological choices. Following is a list of points that should be addressed to improve the clarity of the work.

1) The experimental protocol is not very clear: e.g., how many trials were performed in each session? How apart were the sessions? A figure summarizing the experimental protocol would be of help. For example a timeline representing experimental days and replay sessions for each participant. Also, participants have different experimental days but roughly the same number of created sensations. How many objects-sensation were produced during each session?

2) My main question is how natural were the evoked sensations because all participants provided neutral score to the questionnaire item "I could not create the sensation I wanted". The authors argue that the chosen parameters are much more complex than JND and that the evoked sensations are different from those used in classical stimulation studies. Moreover, replay sessions without visual stimulation provided above chance results. Still, I wonder how the subjects would have responded in replay session in presence of visual stimulation but with an object non-matching the stimulus parameters.

3) Fig 4b: why LDA performance are tested with combinations of parameters always involving amplitude? Why not all combinations are explored?

4) Fig S2: are the data presented representing aggregated data from all sessions? If so, I wonder what would happen with single session data. Would the result be the same or vary across days? Maybe this is not a viable analysis because of data paucity in single session but I am not sure (see point 1).

5) Since many results are presented as aggregated data across all sessions (i.e., fig 2, 3, 6, 8) I wonder whether these are indeed stable in time or may change. The authors make the point that since at the beginning of each session parameter combinations are randomized, learning is prevented but I wonder if some sort of learning, at least in the procedure, may have occurred. Were the subjects spending less time in exploring the parameters across days/sessions? were they truly looking for optimal natural-resembling sensation or looking for previously evoked sensations (see point 2)?

Minor points:

- Fig 2b is missing a title for left panels, I guess "cat".
- Fig S1a should have shaded area for std.
- Fig 5: what is the range? is it normalized?
- Figure labels could be more explicative, e.g. "performance" or "standard deviation" are not immediately intuitive as it is not clear what they are referred to.
- Page 9 line 360: how much is the high frequency? What are the preferred frequencies for the last participant?
- Page 11 line 460 needs rephrasing.
- I do not get the Bonferroni correction for the post-hoc test, why 10?
- Fig 7: was the test not performed for participant P3 or it is simply not reported? Also, is it possible that poor performance of P3 in the replay task is due to the fact that this session was 17 weeks apart from the last experimental session?

Version 1:

Reviewer comments:

Reviewer #1

(Remarks to the Author)

The authors have address my concerns in their revision.

(Remarks on code availability)

The provided code is sufficient

Reviewer #2

(Remarks to the Author)

The authors have adequately addressed my concerns.

(Remarks on code availability)

Reviewer #3

(Remarks to the Author)

The authors have addressed all the points raised in my previous report.

(Remarks on code availability)

November 27th, 2024

Topic: Response to reviews
Manuscript number: NCOMMS-24-17014-T

We are grateful to the reviewers for their critical and helpful comments. To address the reviewers' concerns, we have conducted additional thorough and more objective analyses of our data. With these novel results, we underline the robustness and importance of our study. Please find enclosed our revised manuscript, where edits are highlighted in yellow. We address each reviewer's comments in our point-by-point response below. Whenever we use line and page numbers in this response, we refer to those of the revised manuscript.

Reviewer Comments

Reviewer #1:

In this manuscript Verbaarschot, Gaunt and colleagues demonstrate a novel paradigm for individuals with bi-directional intracortical brain-computer interfaces to explore the stimulation parameter space. This is an important and timely issue as generating robust sensory responses from intra-cortical microstimulation parameters is key in providing usable feedback to improve prosthetic control. The method shows promise, however, more is required to ensure that these results are robust given the reliance on the subjective experience of a small sample.

We thank Reviewer #1 for their recognition of the importance and novelty of our work and will address each major and minor concern below.

Major:

My biggest concern in this manuscript is the reliance on the subjective experience of a small sample size to demonstrate that this method is effective in improving the sensory experience of individuals using bi-directional intracortical brain-computer interfaces. This issue is exemplified in the fact that subject P3 spends the least time exploring the parameters space, has no significance in the parameter selection fitting, yet has a higher satisfaction than P2, more similar to that of the highest respondent C1. This indicates that relatively arbitrary parameters, identified quickly could be enough for patients to appreciate, which contradicts the sentiment of the manuscript. This confound should be better discussed in the manuscript. I appreciate that there are very few participants at this stage able to perform experiments such as these, however given this, more objective analysis would be expected, or more participants should be included in such a study before interpreting the results.

As Reviewer #1 recognizes, the sample of available participants for this type of research is limited. Across the globe, there were about seven participants with bi-directional intracortical implants in their somatosensory and motor cortex available at the time of this study. Three of these participants were included in this study, two at the University of Pittsburgh and one at the University of Chicago. Previous research on bi-directional BCIs almost exclusively reports results of a single participant (e.g., Armenta Salas et al., 2018; Fifer et al., 2022; Flesher et al., 2016; Flesher et al., 2021; Osborn et al., 2021). Having three participants rather than one, is a uniquely large sample in this type of research. Although having an even larger participant sample would be highly desired, extending the sample beyond this number was simply not possible at the time. We have clarified the restriction of our sample size in the Discussion section:

“The sample of available participants for this type of research is limited. Across the globe, there were about seven participants with bi-directional intracortical implants in their

somatosensory and motor cortex available at the time of this study. Three of these participants were included in this study, which is a uniquely large sample compared to previous research^{8-10,16,27}. Extending the sample beyond this number was simply not possible at the time.” (Discussion, page 8, lines 322-326)

The goal of our experiment was not to find satisfying sensations, but sensations that were appropriate for different object interactions (page 3, lines 76-78). If the participant cannot associate object-specific tactile characteristics with their created sensations, they offer little informational value. Despite our encouragements, participant P3 was less extensive in his exploration of the parameter space compared to P2 and C1 (Fig. 2b, page 22). If a participant does not explore the full parameter space on each trial, they do not know all possible options and cannot make an informed choice. The absence of a significant relation between the chosen stimulation settings and objects, and the participant’s inability to recognize what object a previously created sensation belonged to shows that such a quick search strategy was not effective. Since both P2 and C1 were able to select object-specific stimulation settings that evoked recognizable and distinct touch percepts, we believe our method can be effective, provided that the participant explores the full extent of the parameter space.

To provide further credence to the effectiveness of our method, we conducted novel analyses and modified Figures 3 and 4. Figure 3a (page 24) now demonstrates the clear separation between the stimulation parameters that were chosen by participants P2 and C1 for each object. Figure 3b confirms this by showing that the chosen stimulation parameters within a same object category were significantly more similar to each other than those of a different object category. Moreover, Figure 3c (page 24) demonstrates that a participant was more likely to correctly recognize a sensation in absence of its visual identity when the chosen stimulation parameters were closer to the median choice for that object. This suggests that there are unique locations in the parameter space that are associated with the sensory characteristics of a specific object. Figure 4e (page 26) confirms this by showing that participants more easily confuse two objects with each other in the replay task when their associated stimulation parameters are more similar to each other.

We have included the results of our novel analysis in our Results section and have extended our interpretation of participant P3’s results in the Discussion section:

“Although the variance in participant’s parameter selections was high, the resulting stimulus profiles occupied distinct locations in the parameter space, depending on their target object (Fig. 3a). The separation between object-specific stimulus profiles was less clear in participant P3 compared to P2 and C1, likely due to his shortened exploration times (Fig. 2b-d). Nevertheless, the parameter selections of sensations that participants recognized consistently correctly in the replay task, showed greater similarity within the same object category than across different objects (P2: $p = 0.005 \cdot 10^{-3}$, one sided Wilcoxon rank sum, $\alpha = 0.05$, $z = -4.428$; C1: $p = 0.004 \cdot 10^{-1}$, $z = -3.362$, Fig. 3b), albeit non-significantly for P3 ($p = 0.084$, $z = -1.379$, Fig. 3b). The closer a stimulus set was to the median across all successfully recognized stimulus profiles of a certain object, the more likely the participant was to correctly identify the sensation as belonging to that object (P2: $p = 0.003 \cdot 2$, one sided Wilcoxon rank sum $\alpha = 0.05$, $z = -4.146$; P3: $p = -0.007 \cdot 1$, $z = -3.396$; C1: $p = 0.011$, $z = -2.532$, Fig. 3c).” (Results, page 4, lines 130-140)

“In general, we observed that the more distinct the chosen stimulus profile was, the easier participants could distinguish them in the replay task (Pearson’s $r(18) = -0.45$, $p = 0.045$, Fig. 4e)” (Results, page 5, lines 171-173)

“Participants P2 and C1 selected distinct stimulation profiles per object (Fig. 3) that evoked recognizable and distinct touch percepts in the absence of a visual context (Fig. 4). Like P2 and C1, participant P3 was very satisfied with his created sensations (Fig. 6a). However, for most sensations he was unable to correctly recognize what object they belonged to in the replay task (Fig. 4a,d). His chance-level performance can be explained by insufficient exploration in the object-sensation mapping task (Fig. 2b-d). If a participant does not explore the full parameter space on each trial, they cannot make an informed choice. The absence of a significant relation between the chosen stimulation settings and objects, and the participant’s inability to recognize what object a previously created sensation belonged to shows that his quick search strategy was not effective. In contrast, the object-specific stimulus selections of P2 and C1 show that our method can be effective, provided that the participant explores the full extent of the parameter space.” (Discussion, page 7, lines 275-285)

Armenta Salas, M. *et al.* Proprioceptive and cutaneous sensations in humans elicited by intracortical microstimulation. *eLife* **7**, e32904 (2018).

Fifer, M. S. *et al.* Intracortical Somatosensory Stimulation to Elicit Fingertip Sensations in an Individual With Spinal Cord Injury. *Neurology* **98**, (2022).

Flesher, S. N. *et al.* Intracortical microstimulation of human somatosensory cortex. *Sci. Transl. Med.* **8**, (2016).

Flesher, S. N. *et al.* A brain-computer interface that evokes tactile sensations improves robotic arm control. *Science* **372**, 831–836 (2021).

Osborn, L. E., Christie, B. P., McMullen, D. P., Nickl, R. W., Thompson, M. C., Pawar, A. S., ... & Fifer, M. S. (2021, November). Intracortical microstimulation of somatosensory cortex enables object identification through perceived sensations. In *2021 43rd annual international conference of the IEEE engineering in medicine & biology society (EMBC)* (pp. 6259-6262). IEEE.

The methodology proposed for participants to use shows promise, however there are controls that should be implemented to ensure the robustness of the results presented. There is no mention of catch trials anywhere in the manuscript, which are important to show that the evoked sensations are related to the changing stimulus parameters themselves and not evoked purely based on the expectation of a response.

We agree with Reviewer #1 that including catch trials in the replay task, in which we would present a random set of stimulus parameters rather than one that was selected by the participant, would be a good addition to this experiment. The replay performance on these catch trials would serve as a baseline to compare the other sensations against; whereas we do not expect randomly created sensations to show a consistent link to a particular object, participant-selected sensations should correlate with particular object choices. To indicate this limitation of our study design, we have included this potential analysis in our Discussion section:

“One way to strengthen the replay task would be to include catch trials; trials that present a random stimulus profile. The replay performance on these catch trials would serve as a baseline to compare the other sensations against; whereas we do not expect randomly created sensations to show a consistent link to a particular object, participant-selected sensations should correlate with particular object choices.” (Discussion, page 8, lines 343-347)

However, even absent catch trials, we believe our results stand. If our new methodology was not effective at identifying object-appropriate sensations, we would expect participants to select random stimulation parameters that show no clear differences between objects. If this was true, the participant's replay performance and the LDA performance would be random, as there would be no clear relation between the selected stimulation parameters and target object. Our results show the opposite: in two out of three participants, we see a significant relation between the selected stimulation parameters and object identity (Fig. 3a-c, page 24). This relation is demonstrated through the participant's significant replay performance and the significant LDA performance (Fig. 4a,b, page 26). The significant replay performance informs us that participants can recognize what object a previously created sensation belongs to without seeing any visual cues on the object's identity. The only thing participants can base their choice on is their perception of the chosen stimulation parameters. Moreover, the significant LDA performance shows that there is a higher variability in stimulation parameters across objects compared to within objects. In other words, there is a consistency and distinctiveness in the object-specific stimulation parameters across sessions.

Following from the previous point, although parameter axes are shuffled and blinded it still seems that significant portions of the parameter space are not explored. Was there any effort made to initialize the participants in different parts of the parameter space? This could indicate that participants fall into local minima in which they are able to interpret the sensations in a way they find satisfying. To address this, the authors may consider performing the experiments again, but sequentially 'leaving out' parameters that participants find satisfying without participants knowing. Either participants will always be able to evoke useful precepts meaning that this method cannot overcome the power of sensory expectation and does little to usefully constrain the stimulation parameter space, or participants will eventually reach a parameter space where they are unable to find any parameters for useable sensations in which case the methods presented will be demonstrated more robustly.

The cursor in both explorative parameter spaces always started at the same point: $x = 0$ and $y = 0$. The four stimulus parameters were randomly assigned to the X and Y axes of each rectangle on each trial. In addition, the progression of each parameter was randomly flipped on each trial, meaning that, e.g., $x = 0$ could correspond to either the highest or lowest level of that parameter. Although the cursors always started from the same visual position, the corresponding stimulus parameters were different on each trial due to this random assignment of both parameter type and progression along the rectangle axes. In addition to these random start positions, we highly encouraged participants to take their time to fully explore the parameter space.

Two of the tested participants, P2 and C1, reported clear strategies with which they explored the parameter space. Participant P2 tried the most extreme parameters first, corresponding to the corners of each rectangle, and then continued his search from the most applicable corner (Fig. 2a, page 23). Participant C1 started from the middle of the parameter space, which is roughly identical across all trials as it is the middle level of each parameter. He then moved the cursor in each possible direction (i.e., up, down, left, right) and chose to pursue the most appropriate sensation direction in more detail (Fig. 2a, page 22). Although both approaches may not explore each individual stimulus setting, they do sample the full parameter space first, after which zooming in to a more specific area. The effectiveness of their exploration strategies is strengthened by the object-specificity of their parameter selections (Fig. 3a-d, page 24): the significant LDA classifications of stimulus parameter selections indicate that there is a consistency with which

participants select object parameters across sessions, and that these selections are significantly different for each object (Fig. 4a,b, page 26).

We agree that it is possible that our participants find local minima in their exploration of the parameter space. Perhaps this explains why there is quite some variation within object-specific parameter selections (Fig. 3a, page 24). Theoretically, the proposed control experiment of Reviewer #1, in which we would gradually delete preferred parameter selections from the exploration space, is very interesting. However, we do not believe that this task is practically feasible. The total number of possible stimulus combinations is $10^4 = 10.000$. Deleting stimulus combinations one by one until participants are unable to make a selection would take a very long time. We expect that the repetitive nature of this task will fatigue the participants, which is what we tried to avoid with our experimental design. Moreover, instructing participants to find the most appropriate stimulus setting for a given object would always return a result. However, as you delete stimulus parameters from the space, this result would become more and more random, and less object-specific. Given that the parameter selections of participants P2 and C1 are significantly distinct per object (Fig. 4, page 26), this already shows that there is a valid relation between the selected stimulus parameters and target objects.

We have clarified these points in our Discussion section:

“It is possible that our participants found local minima in their exploration of the parameter space, which could explain the high variability within object-specific parameter selections (Fig. 3a). However, the parameter selections of participants P2 and C1 were significantly distinct per object (Fig. 3), showing a valid relation between the selected stimulus parameters and target objects across sessions. This was confirmed by the significant LDA performance, showing a higher variability in stimulation parameters across objects compared to within objects (Fig. 4a,d).” (page 9, lines 348-353).

The authors here attempt to constrain a large parameter space when considering stimulation evoked sensations. However, by adding 5 more objects they in fact increase the parameters space problem 5-fold! Although significant, the classification accuracy of the LDA model seems low and potentially the significance is driven by the low chance level achieved by having multiple classes. The authors may consider a confusion matrix showing of the significance of the LDA performance if only pairs of classes are tested.

The chance level with five objects is at 20% for the sensation-object mapping (replay) task. We have conducted careful tests (page 14, lines 571-584, lines 600-610) to ensure that our results are well above the statistically valid significance levels, which is higher than the theoretical chance level. We appreciate Reviewer #1's helpful suggestion to investigate the LDA performance across different subsets of objects and have added a panel to Figure 4 (Fig 4a, page 26). This panel shows the classifier and participant performances comparing a single object class to all others, e.g., how well both the classifier and participant could recognize an “apple”-sensation compared to all other object-sensations. Participants P2 (cat, key, towel, toast) and C1 (cat, apple, towel) recognize at least 3 individual object-sensations significantly above chance. Participant P3 has only a significant replay performance on key-sensations.

Our paper shows that different subgroups of object-sensations are more distinct than others. We showed that the LDA performance was higher when grouping sensations based on their level of object compliance or temperature rather than their object identity (page 6, lines 234-243). By grouping soft (i.e., cat, towel) and hard (i.e., apple, key, toast) sensations, the significant LDA performances increased to 63-71% for participants P2 and C1, respectively. Similarly, grouping cold

(i.e., key, apple) and warm (i.e., towel, toast, cat) sensations led to significant LDA performances of 72-76% for participants P2 and C1, respectively. These increases in performance are also reflected in the replay data of P2 and C1. When grouping sensations based on their compliance level, participants P2 and C1 reached a significant replay performance of 66-81%, respectively. Similarly, P2 and C1 reached a significant replay performance of 65-76% by grouping sensations according to their temperature level. Although the LDA classifiers did not reach a significant performance for different levels of friction or texture (smooth: cat, apple, key; rough: towel, toast), the replay performances of P2 and C1 did, with significant performances of 58%-65%, respectively. Lastly, participant C1 reached a significant replay performance of 63% by grouping sensations based on their macro structure (round: cat, apple, towel; edged: key, toast). These results indeed confirm that different groups of object sensations can achieve higher performance results. We have added the novel replay results on different combinations of sensations to our Results section:

“Similar results were obtained by re-assessing the participant’s replay performance using the same compliance and temperature labels as used for the LDA classifications (Fig. 7d,e). Based on different levels of compliance, Participants P2 and C1 reached an average replay performance of 66-81% (P2: $p = 0.001$, one-sided permutation test, 1000 permutations, $\alpha = 0.013$; C1: $p = 0.001$), respectively. Similarly, P2 and C1 reached an average replay performance of 65-76% by grouping sensations according to their temperature level (P2: $p = 0.001$; C1: $p = 0.001$). Although the LDA classifiers did not reach a significant performance for different levels of friction or texture (smooth: cat, apple, key; rough: towel, toast), the replay performances of P2 and C1 did, with average performances of 58%-65%, respectively (P2: $p = 0.010$; C1: $p = 0.001$). Lastly, participant C1 reached an average replay performance of 63% (P2: $p = 0.025$; C1: $p = 0.001$) by grouping sensations based on their macro structure (round: cat, apple, towel; edged: key, toast). These results suggest that participants attempted to design intuitive sensations, as the similarities between stimulus profiles reflected the expected tactile similarities of their target objects.” (Results, page 6, lines 244-255).

In addition, we have conducted novel analyses and edited Figure 7 accordingly. Figure 7a demonstrates a positive correlation between any two objects that participants confuse with each other in the replay task, and the tactile similarity between them (page 30). The tactile similarity is based on the survey ratings of a group of people with intact somatosensation (Fig. S1). As the tactile similarity between two objects increases, people are more likely to confuse the created sensations for these objects. For example, apples and keys are both cold, smooth and hard. The created sensations for these objects reflect their similarities, as their selected parameters are more similar and the resulting sensations are more often confused with each other (Fig. 7d,e, page 30). Figure 7b shows that, for all participants, the chosen stimulation parameters are more similar within similar levels of object compliance than across different ones. The same holds for object temperature. These results are now included in our revised paper:

“The stimulus parameters that our participants selected correlated with these expected tactile characteristics; the more tactile characteristics two objects shared, the more likely they were confused by our participants in the replay task (Pearson’s $r(28) = 0.30$, $p = 0.104$, Fig. 7a). Specifically, significant differences in compliance ($p = 0.001-25$, one-sided Wilcoxon signed rank test, Bonferroni corrected at $\alpha = 0.013$, $z = -11.055$) and temperature ($p = 0.004-54$, $z = -15.884$) explained most of the observed variance between different object groups (Fig. 7b, Fig. S9). For example, soft (i.e., cat, towel) and hard (i.e., apple, key,

toast) object sensations were much more like sensations in that same compliance category than the opposite one.” (Results, page 6, lines 225-232)

In order to utilize bi-directional brain-computer interfaces over time and minimize training time, a lot of effort recently has been placed on the ability to decode longitudinally and across participants. The same could be true for evoking sensations. The authors discuss a poor longitudinal stability for the stimulation parameters, but I would be interested if the authors looked at training on one participant and testing on the others to see how stable the parameter selection is across participants.

We agree with Reviewer #1 that it would be very interesting to determine whether the chosen stimulus parameters could transfer across people. It is not trivial to assume that the stimulus parameters that work for one participant will transfer to another. In case they do not, there is a multitude of reasons that may explain this. Differences in brain anatomy, exact location of the microelectrode arrays, location of electrodes that can evoke a reliable sensation, and range of possible stimulation parameters may exist between participants. Accurately determining both the long-term stability of the chosen stimulation parameters and their transfer success across participants requires a dedicated experiment.

We have added Figures 3d (page 24) and 7c (page 30) to provide a better view of the common stimulus parameters that were selected per object and tactile characteristic. One could see these figures as stimulus recipes that can be used to evoke appropriate sensations for different object interactions. We have included these novel results in our Results section:

“From these successfully recognizable stimulus profiles, a rough mapping from stimulus parameters to sensory object characteristics could be inferred (Fig. 3d).” (Results, page 4, lines 140-141)

“As such, a rough mapping could be identified between stimulus parameters and different levels of perceived object compliance and temperature (Fig. 7c).” (Results, page 6, lines 232-233)

Whether this mapping would indeed be effective needs to be tested in future research.

Minor:

Line 40: ‘In the few human studies that exist, participants reported sensations like “buzzing”, “tingle”, and “pressure” in response to simple stimulation’. I do not believe this statement fairly reflects the broad range of naturalistic sensations captured in references 8,10,16.

We have added a more careful and encompassing summary of previously reported sensations in our Introduction:

“Human participants typically describe ICMS-evoked sensations as “possibly natural”¹⁰, reporting a wide variety of naturalistic qualities such as “pressure”, “press”, “tap”, “warm”, “squeeze”, “pinch”, “vibration”, “blowing”, and “goosebumps”^{8,10,16}. In addition, some more artificial qualities such as “pins and needles”, “tingle” and “electrical” have been reported^{8,10,16}. The perceived qualities depend on the stimulation amplitude, frequency and (multi-electrode) location(s)^{8,10,15,17,18}.” (page 2, lines 48-53):

The ‘cat’ label is missing in figure 2b.

We have corrected this error and added the missing cat label to Figure S3.

Reviewer #2:

Verbaarschot and colleagues present a well-executed and interesting set of experiments and results. Their attempt to address questions related to phenomenology and naturalness in artificial somatosensation restoration via ICMS is timely and exciting.

We thank Reviewer #2 for their interest in our study and positive comments.

Major:

If I understand correctly, the user in this protocol used only one hand to define the four parameters. In the current study, it is not clear if they had to switch between two tablets if they fixed the first two parameters and then the last two (or if they switched between the two). Do you think this sequential choice of parameters could affect the results? Would non-sequential (e.g., 3D control + rotation of the left hand - assuming the patient was able to do it) have changed something?

The participants used their left hand to interact with a single tablet interface. As shown in Figure 1a (page 20), this tablet included both rectangle parameter spaces and the target object. Participants needed to interact with the object image to feel a sensation, either by touching the image, or by using an automated cursor that swept back and forth across the object at a regular interval. Participants were free to move either one cursor at a time or both of them before trying out the novel sensation. The division of the stimulus space across two rectangles could have influenced the results. However, the random assignment of stimulus parameters to rectangle axes would prevent a consistent effect of this potentially sequential procedure across trials. Therefore, we believe that it is unlikely that it influenced our main result.

The verbal descriptors reported in Fig 5 are exciting. If I understand correctly, the standard survey is the one the same authors presented in their previous papers published in 2016 and 2021. The broadening of the reported sensations compared to the standard survey is very interesting, but is it possible that it is a simple effect of the presence of the objects that they chose to work with? Do participants 'simply' give new descriptions because they are shown a cat, or is there more to it?

Reviewer #2 is correct, the standard survey is the one that was used in previous publications (Flesher et al., 2016; Hughes et al., 2021) and is based on that of Kim et al. (2018).

As explained in our Discussion (page 8, lines 303-310), we agree with Reviewer #2 that our experimental context could explain the more elaborate sensation reports in comparison with previous research. If you are asked to create a sensation of a cat and you touch a picture of a cat while you experience that sensation, you are likely to say it feels “soft”, “warm” and “hairy”. However, we believe that this is not much different from a person with intact somatosensation who is petting a cat in a real-life situation. In that case, they also see and touch a cat. Through repeated association throughout their life, they have come to experience or describe that particular multi-sensory experience as feeling “soft”, “warm” and “hairy”. The perceptions that are artificially generated and described in our experiment may not be that different. At the very least, our participants seemed convinced of the appropriateness of their created sensations within the given experimental context.

To assess the extent to which the sensory experiences created in our experiment could convey object-appropriate characteristics, we determined the separability of the selected stimulation

parameters and their perceptual distinctiveness per object or tactile characteristic. As described in the Results section, the chosen object-specific stimulation parameters showed a significant separation between different objects (page 4, lines 130-149, Fig. 3a-d, Fig. 4a,d). In addition, participants P2 and C1 were able to recognize to which object a sensation belonged to with significant accuracy in absence of any visual context during the sensation-object mapping (replay) task (page 4, lines 158-164, Fig. 4a,d). Interestingly, the confusion between two sensations increased as the tactile characteristics of these objects were more similar (new Fig. 7a, page 30). Here, object similarity was based on the survey results of a group of people with intact somatosensation. Differences in both object compliance and temperature explained perceived differences in the created sensations and their corresponding stimulation parameters (new Fig. 7b,d,e, page 30). These results only show that the Euclidean distance between different sensations correlates with the expected tactile similarity of the corresponding objects (Fig. 4e on page 26, Fig. 7a on page 30). This does not mean that the created sensations necessarily felt like something cold, warm, soft or hard. However, in combination with the congruent vivid sensation reports (Fig. 5, page 28), these results do suggest that the created sensations contained some intuitive tactile resemblance to their target object.

We have elaborated our discussion of the effect of our particular experimental context on our Results section:

“The difference in experimental context could explain why the participants in this study spontaneously reported novel and more object-oriented sensation descriptors (Fig. 5). If you are asked to create a sensation of a cat and you touch a picture of a cat while you experience that sensation, you are likely to say it feels “soft”, “warm” and “hairy”. However, we believe that this is not much different from a person with intact somatosensation who is petting a cat in a real-life situation. In that case, they also see and touch a cat. Through repeated association throughout their life, they have come to experience or describe that particular multi-sensory experience as feeling “soft”, “warm” and “hairy”. The perceptions that are artificially generated and described in our experiment may not be that different. At the very least, our participants seemed convinced of the appropriateness of their created sensations within the given experimental context (Fig. 5, 6a). As people will ultimately use closed-loop brain-computer interfaces in daily life, it is important to investigate the functional use and experience of ICMS-evoked sensations in their relevant multi-sensory context.” (page 8, lines 308-320)

Flesher, S. N. *et al.* Intracortical microstimulation of human somatosensory cortex. *Sci. Transl. Med.* **8**, (2016).

Hughes, C. L. *et al.* Perception of microstimulation frequency in human somatosensory cortex. *eLife* **10**, (2021).

Kim, L. H., McLeod, R. S. & Kiss, Z. H. T. A new psychometric questionnaire for reporting of somatosensory percepts. *J. Neural Eng.* **15**, (2018).

Minor:

Line 51: References 8, 10, 17, and 18 seem misplaced. Are the authors saying that all these experiments provided engaging experimental paradigms?

We thank Reviewer #2 for pointing this out. The references are meant to support the opposite: experiments that require the participant to fill out a survey after each experimenter-driven stimulus manipulation. We have changed the sentence to:

“Moreover, through the self-paced scanning of a pre-selected parameter space (Fig. 1b), participants could adjust stimulation parameters without having to complete a sensory survey after each new parameter^{8,10,17,18}, providing an efficient and engaging experimental paradigm.” (page 2, lines 62-65).

Introduction: Can you please describe the challenges in terms of the 'phenomenology' of the perceived feedback when using ICMS?

A first challenge in phenomenology is capturing subjective experiences into objective measurements. For example, we cannot guarantee that two experiences are similar even when they are described in exactly the same words or are given exactly the same ratings on a survey. Moreover, the way in which we ask about these experiences can determine the outcome. If I ask a person how much “pressure” they felt on their hand during a certain stimulus, a participant may interpret a “hard” sensation feeling like high pressure and a “soft” sensation as low pressure. However, they may not have described such sensation as feeling like “pressure” in absence of this survey question. In addition, subjective reports can be sensitive to bias. If I ask a participant how “natural” a sensation feels they may figure out that I am trying to create naturalistic sensations and answer in favor of that goal. Besides these more theoretical challenges, there are also practical challenges; collecting verbal reports or survey ratings takes time and can feel very repetitive.

In this study, we addressed these issues by having the participants create their own sensations. This way, participants do not depend on an experimenter and can repeatedly judge and adjust a sensation without any interruptions.

We have clarified this issue in our Introduction section:

“Moreover, measuring artificial percepts can be challenging due to the inherent complexity of capturing an experienced sensory quality into objective measurements. We cannot guarantee that two experiences are similar, even when they are described with exactly the same words. Sensation reports can be difficult to interpret and prone to bias. For example, if a person is asked how much “pressure” they felt on their hand during a stimulus, a participant may interpret a “hard” sensation as feeling like high pressure and a “soft” sensation as low pressure. However, they may not have described the sensation as “pressure” at all in the absence of this question.” (page 2, lines 39-45)

“Evaluating the resulting sensations one by one using verbal reports or survey ratings may not be effective due to the theoretical challenges described above, and their time consuming and repetitive nature.” (page 2, lines 54-56)

“Moreover, through the self-paced scanning of a pre-selected parameter space (Fig. 1b), participants could adjust stimulation parameters without having to complete a sensory survey after each new parameter^{8,10,19,20}, providing an efficient and engaging experimental paradigm.” (page 2, lines 62-65)

Line 108: What is the difference between time exploring and the amount of stimulation?

To adjust stimulation parameters, participants changed the location of the red cursors in the blue and green rectangle spaces (Fig. 1a). To experience the resulting stimulus, participants either touched the object that was depicted at the top of the screen, or activated an automated cursor that swept back and forth across the object at a regular time interval. As such, participants received stimulation during only a subset of the total trial time. Therefore, the time to trial completion was

not the same as the total amount of time that stimulation was provided during a trial. We have clarified this in our Methods and Results section:

“The participants received stimulation during only a subset of the total trial time; whenever they interacted with the presented object via their own hand or the automated cursor.” (Methods, page 12, lines 483-485).

“Participants P2 and C1 had similar trial completion times (Fig. 2b), explored roughly equal parts of the parameter space (Fig. 2c) and received comparable amounts of stimulation during a trial (Fig. 2d). In contrast to P2 and C1, participant P3 spent significantly less time on each trial, exploring less of the parameter space and receiving less stimulation (Fig. 2b-d).” (Results, page 4, lines 126-129)

Could the difference in score be explained by the somatotopic projection of the three stimulated areas? Could that explain the rather poor results of P3 (for whom the blue and orange stimulation points are more overlapping)?

Since this research study focuses on the quality of the sensation, rather than the location, we do not think this can explain the results.

In Fig 2B, the label 'cat' is missing.

As indicated in our response to Reviewer #1 above, we have corrected this error (Fig. S3).

Please follow the same order of materials as much as possible throughout the figures.

We have made sure to refer to each object and stimulus parameter in the same order throughout our figures. We have also double checked the order in which each figure is referenced in the text.

Curiosity: What justifies the choice of n=34 for the healthy participants group?

The target group size of our survey study was actually set to 30 participants, as we believed that to be a sufficiently large sample to get a sense of the tactile characteristics that are generally associated with different objects. Due to a very smooth and fast data collection process, we ended up with slightly more participants.

Lines 192-198: I did not understand how you get the ground truth here. How do you find the values for the orange and blue circles? Where in the replay or the LDA is the information about hardness and temperature?

The labels are based on the object quality survey results (page 6, lines 220-225). In other words, what, e.g., compliance and temperature levels did a group of 34 people with intact somatosensation ascribe to the objects used in this study? From this data, we inferred that, e.g., cats are generally perceived to be soft and warm, whereas keys are perceived to be cold and hard (Fig. S1). We then re-labeled our data using these survey-informed labels. For example, for the “compliance” analysis, we grouped soft (cat, towel) and hard (apples, keys, toast) objects into two different groups. With these new labels, we then re-calculated the performance of our participants on the replay task, and re-trained an LDA classifier to predict these tactile characteristics rather than object identities. We have edited Figure 7d,e to better visualize the result (page 30) and clarified this procedure in the Results section:

“To further assess the correlation between stimulus profiles and expected tactile object characteristics, individual LDA classifiers were trained to make predictions on different levels of object compliance, temperature, friction/texture and macro structure based on sets of stimulus parameters. To do so, the data was relabeled using the quality survey results. For example, for the “compliance” analysis, soft (cat, towel) and hard (apples, keys, toast) objects were grouped into two different groups. A novel LDA classifier was then re-trained to predict these tactile characteristics rather than object identities. This analysis confirmed the results above: for both P2 and C1, binary levels of object compliance (P2: 63±12% accuracy, $p = 0.010$, permutation test with 1000 permutations, $\alpha = 0.025$; C1: 71±14% accuracy, $p = 0.002$, Fig. 7d) and temperature (P2: 72±11%, $p = 0.001$ C1: 76±13%, $p = 0.001$ Fig. 7e) could be significantly predicted. LDA classifier performance was not significant for the other tactile characteristics (Table S2).” (page 6, lines 234-243)

Did the participants have residual sensation in their hand (in particular C1)? How did they relate the phenomenological experience in real life versus via ICMS?

All participants have residual sensations in their hand. From all participants, C1 has the most preserved sensation. However, all participants describe their residual sensation as feeling non-natural, like tingles. In contrast, all participants described vivid memories and ideas of what the target sensations in this experiment should feel like. They had no doubt or confusion about what sensation they wanted to achieve for the various object interactions. We have clarified this in our Methods and Results section:

“All participants had some residual sensations in their right hand, although they described these sensations as feeling non-natural.” (Methods, page 10, lines 386-387).

“All participants described vivid memories and ideas of what the target sensations in this experiment should feel like. They expressed no doubt or confusion about what sensation they wanted to achieve for the various object interactions.” (Results, page 4, lines 119-121)

In general, participants describe sensations evoked via intracortical microstimulation of the somatosensory cortex as “possibly natural” (93.2%) to “almost natural” (4.8%) (Flesher et al., 2016). Changes in stimulus parameters can influence the perceived naturalness of these sensations. For example, where 20 Hz stimulations predominantly evoke naturalistic sensations of “pressure” (50%), “tapping” (32%), “touch” (26%) and “sparkle” (16%), 300 Hz stimuli tend to evoke more artificial sensations of “tingle” (72%), “warm” (31%), “sharp” (11%), “buzzing” (9%) and “vibration” (8%) (Hughes, Flesher, Weiss, Boninger, et al., 2021). A recent study directly compared sensations evoked by mechanical stimulation of the skin to those evoked by intracortical microstimulation of the somatosensory cortex (Hobbs et al., 2024). This study confirms that the choice of stimulus parameters is crucial to the naturalness of the perceived sensation, where biomimetic paradigms (those that aim to mimic the neural response to actual touch) feel more like actual touch traditional stimulation approaches.

Flesher, S. N., Collinger, J. L., Foldes, S. T., Weiss, J. M., Downey, J. E., Tyler-Kabara, E. C., ... & Gaunt, R. A. (2016). Intracortical microstimulation of human somatosensory cortex. *Science translational medicine*, 8(361), 361ra141-361ra141.

Hobbs, T. G., Greenspon, C., Verbaarschot, C., Valle, G., Boninger, M., Bensmaia, S. J., & Gaunt, R. A. (2024). Biomimetic stimulation patterns drive natural artificial touch percepts using intracortical microstimulation in humans. *medRxiv*, 2024-07.

Hughes, C. L., Flesher, S. N., Weiss, J. M., Boninger, M., Collinger, J. L., & Gaunt, R. A. (2021). Perception of microstimulation frequency in human somatosensory cortex. *Elife*, *10*, e65128.

Reviewer #3:

The authors present a study on sensory stimulation of participants with tetraplegia by means of invasive brain electrodes. The main novelty of the study is that participants were able to choose the stimulation parameters for different objects after an exploration phase.

The authors support their results with many tests and motivate their findings with different analyses. However, while the manuscript is overall well written and enjoyable to read, it is sometimes hard to get the reasons behind some methodological choices. Following is a list a points that should be addressed to improve the clarity of the work.

We thank Reviewer #3 for their time and critical comments to our manuscript. We will address each point in our reply below.

Major:

1) The experimental protocol is not very clear: e.g., how many trials were performed in each session? How apart were the sessions? A figure summarizing the experimental protocol would be of help. For example a timeline representing experimental days and replay sessions for each participant. Also, participants have different experimental days but roughly the same number of created sensations. How many objects-sensation were produced during each session?

People with tetraplegia deal with frequent and unpredictable health issues. For these reasons, scheduled test sessions sometimes needed to be adjusted on the spot. Depending on the participants' physical and mental state, test sessions could sometimes run longer or had to be shortened. We have clarified this in our Methods section:

“Each object was repeated two to three times per session in the object-sensation mapping task. Participant P2 typically created three sensations for three different objects per session. Each session, three out of five objects were randomly selected. P2 thus completed nine object-sensation mapping trials per session. In contrast, participants P3 and C1 created two sensations for all five objects per session, summing to ten object-sensation trials per session. However, people with tetraplegia deal with frequent and unpredictable health issues. For these reasons, scheduled test sessions sometimes needed to be adjusted on the spot. Depending on the participants' physical and mental state, test sessions could sometimes run longer or had to be shortened. Across 22 sessions, participant P2 created 30 cat, key and toast sensations, 33 apple sensations and 29 towel sensations (Fig. S2). Across 10 sessions, participant P3 created 20 cat and toast sensations, 18 apple and towel sensations, and 19 key sensations. Similarly, participant C1 created 21 cat, 19 apple, key and towel sensations and 20 toast sensations across 10 sessions (Fig. S2).” (page 12, lines 502-512).

“Each sensation that was created earlier during that session in the object-sensation mapping task, was tested at least twice in the replay task. Therefore, the replay task

typically consisted of 18 trials for P2, and 20 trials for P3 and C1 (Fig. S2b).” (page 13, lines 519-521)

To further clarify the data collection process across the different sessions for each participant, we have added a new figure to our supplementary materials, showing the data collection timeline for each participant (Fig. S2a) and the number of collected trials per session (Fig. S2b).

2) My main question is how natural were the evoked sensations because all participant provided neutral score to the questionnaire item "I could not create the sensation I wanted". The authors argue that the chosen parameters are much more complex than JND and that the evoked sensations are different from those used in classical stimulation studies. Moreover, replay sessions without visual stimulation provided above chance results. Still, I wonder how the subjects would have responded in replay session in presence of visual stimulation but with an object non-matching the stimulus parameters.

Due to the unnatural activation of large pools of neurons using intracortical microstimulation, and the limited set of stimulation parameters used in this study, we are not surprised that the participants responded neutrally to the statement “I could not create the sensation I wanted”. However, they could also have felt incapable of creating anything suitable for the virtual object interactions, which would have led to a negative answer. In contrast, participants were positive about their created sensations (Fig. 6a, page 29), reported vivid and object-specific tactile characteristics in response to virtual touch (Fig. 5, page 28), and were able to correctly assign created sensations to their target object in absence of a visual context (Fig. 4a,d, page 26), suggesting that they were creating object-appropriate sensations.

We agree with Reviewer #3 that it will be very valuable to further disentangle the influence of visual context on the perceived tactile sensation. Deliberately manipulating the congruency of the visual context with a presented sensation will indeed allow us to do so. However, such an investigation is outside of the scope of our current research and requires a dedicated experiment.

3) Fig 4b: why LDA performance are tested with combinations of parameters always involving amplitude? Why not all combinations are explored?

We have tested all 15 possible combinations of stimulation parameters but chose to display only a subset of them in Figure 4b for simplicity. We have added figure S5a-c to our supplementary materials, showing the LDA performance with each parameter combination for each individual participant. We have also clarified this in our Methods section:

“Since there was a total of 15 possible combinations of stimulation parameters (amplitude, frequency, biomimetic factor, drag, amplitude + frequency, amplitude + drag, amplitude + biomimetic factor, ... , amplitude + frequency + biomimetic factor + drag), the significance level was Bonferroni corrected to $0.05/15 = 0.003$.” (page 14, lines 586-589).

4) Fig S2: are the data presented representing aggregated data from all sessions? If so, I wonder what would happen with single session data. Would the result be the same or vary across days? Maybe this is not a viable analysis because of data paucity in single session but I am not sure (see point 1).

We indeed show the aggregated data from the replay task of all sessions in Figure S6. However, our LDA analysis on the total charge per electrode was conducted in a similar fashion to that using the chosen stimulation parameters. Therefore, the LDA did take single trials into account and served to check whether the significance of our main LDA result could be explained by differences

in total charge. We agree with Reviewer #3 that it would be interesting to further investigate differences in parameter selections within and across sessions. However, with only two to three repetitions per object per session, this study may not be best suited to investigate this question.

We have looked at differences in overall variability in the selected object-specific stimulation parameters within and across sessions. We have reported this result in Figure S7. There, we observe that, on average, the variation in object-specific parameter selections is significantly smaller within a session than between sessions. This difference was mostly explained by significant differences in the variability of selected frequencies.

5) *Since many results are presented as aggregated data across all sessions (i.e., fig 2, 3, 6, 8) I wonder whether these are indeed stable in time or may change. The authors make the point that since at the beginning of each session parameter combinations are randomized, learning is prevented but I wonder if some sort of learning, at least in the procedure, may have occurred. Were the subjects spending less time in exploring the parameters across days/sessions? were they truly looking for optimal natural-resembling sensation or looking for previously evoked sensations (see point 2)?*

We agree with Reviewer #3 that some type of learning could take place. Even though participants could not learn the mapping of stimulus parameters to rectangle axes, they likely get a better feeling over time of what type of sensations are possible to create. To clarify: we randomized the mapping of stimulation parameters to rectangle axes on every single trial. In addition, the progression of each parameter was randomly flipped on each trial, meaning that, e.g., $x = 0$ could correspond to either the highest or lowest level of that parameter.

We did not see a significant change in the total time that participants took to complete an object-sensation mapping trial (see figure below).

We do believe that participants were looking for optimal natural-resembling sensations (page 7, lines 286-299), as confirmed by their verbal descriptions of their created sensations (Fig. 5, page 28) and the correspondence between the relative distance in stimulus parameters and differences in tactile object characteristics (Fig. 7, page 30). Specifically, the extent of differences in object-specific parameter selections reflected differences in the compliance and temperature of those objects (Fig. 7b). This was also apparent from the participant's performance on the replay task: objects that shared similar tactile characteristics were more often confused with each other (Fig. 7a, d, e).

Minor:

Fig 2b is missing a title for left panels, I guess "cat".

As indicated in our response to Reviewer #1 above, we have corrected this error (Fig. S3).

Fig S1a should have shaded area for std.

We happily accommodate the request and have added a the 25-75% ranges around the medians of the data presented in Figure S1a.

Fig 5: what is the range? is it normalized?

The range is shown on the bottom right of Figure 5 (page 28) and runs from 0 to 100%. The shaded areas indicate the percentage of times that these descriptors were used out of all repetitions of that object in the object-sensation mapping trials. This data is not normalized.

Figure labels could be more esplicative, e.g. "performance" or "standard deviation" are not immediately intuitive as it is not clear what they are referred to.

We have clarified the figure labels of Figures 4b (page 26), and 8a and b (page 32).

Page 9 line 360: how much is the high frequency? What are the preferred frequencies for the last participant?

In the frequency intensity-mapping task, the frequency was set to 20 Hz, 85 Hz or 150 Hz. We have clarified these parameter settings in our manuscript (page 11, lines 439-443). We are not sure which participant Reviewer #3 refers to as the "last" one. We describe the results for all participants on page 10, lines 439-443:

"Although participant C1 consistently rated a high frequency (150 Hz) stimulus as most intense on all electrodes, the results of participants P2 and P3 were mixed. Participant P2 seemed to have a low frequency (20 Hz)-preferring electrode, a high frequency (150 Hz)-preferring one and a medium frequency (85 Hz) -preferring one. Participant P3 seemed to have two high frequency (150 Hz)-preferring electrodes and one medium frequency (85 Hz) -preferring one."

This result is also displayed in Figure S11b.

Page 11 line 460 needs rephrasing.

We appreciate Reviewers #3 notice of this error. We have corrected the sentence to:

"Only trials that had a normalized satisfaction score of at least 50 out of 100 were included in our analysis." (Page 13, line 550-551)

I do not get the Bonferroni correction for the post-hoc test, why 10?

There are a total of 10 possible object-combinations that we test: cat-apple, cat-key, cat-towel, cat-toast, apple-key, apple-towel, apple-toast, key-towel, key-toast, towel-toast. We have clarified this in our Methods section on page 13, lines 555-556:

"...testing each possible combination of two objects: cat-apple, cat-key, cat-towel, cat-toast, apple-key, apple-towel, apple-toast, key-towel, key-toast, towel-toast. The Bonferroni-corrected statistical significance level was set to $0.05/10 = 0.005$."

Fig 7: was the test not performed for participant P3 or it is simply not reported? Also, is it possible that poor performance of P3 in the replay task is due to the fact that this session was 17 weeks apart from the last experimental session?

As explained in the legend of Figure 7: “Participant P3 is excluded from this follow-up analysis due to their chance level performance on the replay task” (page 31, lines 925-926). Because we did not find a significant relation between the chosen stimulation parameters and target objects for participant P3, we did not include his parameter selections and replay performances in any further analyses.

To clarify, the sensation-object mapping (replay) task happened at the end of each regular test session (Fig. S2b). This means that on the same day that a participant created a set of sensations, these sensations were also tested in the replay task. We believe that Reviewer #3 may be confused with the two additional replay sessions that were gathered at the very end of our experiment. There was indeed a delay in between those sessions and the last regular test session. However, this data is not displayed in Figure 7 and cannot explain is chance performance on the regular replay task. The chance level performance of participant P3 on the replay task is likely due to the fact that his exploration of the parameter space was less extensive than that of P2 and C1 (Fig. 2b-d, page 22).

We have clarified our methods on page 3, lines 109-117:

“During the first phase (“object-sensation mapping task”, see Methods), participants repeatedly selected stimulation parameters that best represented each object. At the end of each trial, participants were asked to rate their satisfaction with the created sensation. In the second phase, we assessed the discriminability of the created sensations by having participants complete a “replay task” at the end of each session (Fig. 1d). During this task, stimulation trains that were created in the first phase were replayed without the corresponding object image. Participants were then asked to select the object that best matched the experienced sensation. In addition to these same day replay tests, two additional sessions used the replay task to test the discriminability and stability of a selection of sensations that were created across different days. For an overview of the data collection timeline, see Fig. S2.”